# CRAMER: CONTROL VIA REQUEST-AWARE MASKING FOR EDITING RECOMMENDERS

## ABSTRACT

Sequential recommendation models, while powerful, have limited flexibility in responding to immediate user requests, making it difficult to adapt their recommendations to the user's timely interests. Unfortunately, existing user request adaptation methods often incur high computational overhead due to either 1) retraining the entire backbone network or 2) leveraging the inference ability of large language models (a.k.a. prompt engineering), limiting their applicability in large-scale recommendation services. This paper presents **C**ontrol via **R**equest-**A**ware **M**asking for **E**diting **R**ecommenders (**CRAMER**), a framework that takes users' natural-language requests to immediately change sequential recommendation models' behavior. Specifically, inspired by the model control theory, CRAMER treats user requests as control signals to modulate frozen backbone parameters through masking, achieving instant adaptation to diverse requests while avoiding costly retraining. Experiments on multiple large-scale benchmark datasets show that CRAMER outperforms four state-of-the-art request-aware baselines across multiple recommendation metrics while achieving minimal overhead. Moreover, the proposed framework exhibits enhanced controllability and cross-domain adaptability, establishing a new paradigm for request-aware sequential recommendation.

## 1 INTRODUCTION

Sequential recommendation models have advanced the state-of-the-art in predicting users' next interactions by modeling temporal patterns in behavior, but they remain inflexible when users issue immediate requests (Li et al., 2023; Ye et al., 2025). Real-world users frequently want recommendations that reflect on-the-fly intents expressed after viewing an initial recommendation list—for example, when a homepage shows generic suggestions and the user responds with feedback such as "I want more exciting games" or "this headphone is poorly designed." Such responsive request issued in reaction to the system's current output rather than as search queries, introduce several challenges beyond conventional sequential recommendation. First, natural-language requests may emphasize or even contradict historical preferences, requiring the model to dynamically balance immediate intent with long-term behavior patterns (Gao et al., 2021; Radlinski et al., 2022). Second, requests often contain rich semantics such as negations, constraints, or fine-grained attribute preferences, which require accurate interpretation and controllable adjustment of recommendations (Jannach et al., 2021; Moradizeyveh, 2022). Finally, real-time adaptation must be achieved efficiently: sequential recommendation backbones are inherently complex, trained and tested models, so request-aware extensions must rely on lightweight, parameter-efficient mechanisms that preserve responsiveness without full fine-tuning (Houlsby et al., 2019; Prottasha et al., 2024; Shao et al., 2025).

To address these gaps, a variety of strategies have been explored. The most direct is to fine-tune the model to bake request signals into the recommender (Li et al., 2023; Hou et al., 2024), which introduce significant computational overhead. Beyond that, existing work has investigated approaches such as augmenting sequences with requests (He et al., 2022), aligning natural-language requests with item representations (Li et al., 2023; Hou et al., 2024), as well as using large language models (LLMs) or prompt-engineering techniques (Liu et al., 2024; Liao et al., 2025; Zhang et al., 2025). Yet these approaches either rely on shallow representations that cannot capture nuanced user intent, or require domain-specific pretraining, fine-tuning, and heavyweight inference that disrupts service efficiency, scales poorly, and incurs high latency and resource costs. Moreover, prompting often

lacks explainability and suffers from uncontrollable performance due to its sensitivity to prompt phrasing and inherent model hallucinations (Sahoo et al., 2024). As a result, these methods trade off immediacy and deployment efficiency for adaptivity, leaving a need for approaches that can rapidly condition recommendations on arbitrary natural-language requests without costly model updates.

We propose **C**ontrol via **R**equest-**A**ware **M**asking for **E**diting **R**ecommenders (**CRAMER**), a lightweight framework that treats a user's natural-language request as a control input and instantaneously modulates a frozen backbone via parameter masking. Drawing inspiration from model control theory (Li & Rush, 2020; Li et al., 2022), CRAMER applies learned masks to the backbone's parameters so the model's behavior is steered toward the requested intent with minimum computational overhead (Wen et al., 2016; Frankle & Carbin, 2019). CRAMER begins by mean-pooling across all token embeddings from the request to derive a faithful representation of the user's immediate request (Mosbach et al., 2020). This vector serves as a raw control signal for request-to-mask adaption. A Gumbel–Top-$k$ step (Kool et al., 2019) then produces a sparse row–column gate vector, which is decomposed into per-matrix row and column gates and converted into entrywise masks applied to the selected matrices of the frozen Transformer-based sequential recommender. For robustness, CRAMER introduces three masking strategies for attention output matrices and feed-forward networks (FFNs) in the Transformer-based backbone. The training objective for the request-to-mask module combines a prediction loss with a KL regularizer that encourages the learned gate distribution with a sparsity prior (details in Appendix A). Empirically, this masking-based control achieves adaptation with minimal computational overhead, outperforms four state-of-the-art request-aware baselines on multiple large-scale benchmarks, offering a practical, scalable paradigm for request-aware sequential recommendation.

## 2 BACKGROUND AND RELATED WORK

Sequential recommendation aims to predict the next interaction from historical sequences of consumed items, capturing the temporal dynamics beyond static profiles (Pan et al., 2024). In the past few years, Transformer-based models have become the dominant paradigm (Fang et al., 2020), among which SASRec (Kang & McAuley, 2018) and BERT4Rec (Sun et al., 2019) are the most representative, consistently serving as strong baselines across diverse scenarios (Zivic et al., 2024).

Yet, current approaches struggle to adapt when users express immediate intent through natural-language requests (Li et al., 2023). In practice, an immediate request can emphasize aspects of prior preferences, or explicitly negate them (Wu et al., 2019; Luo et al., 2020), thereby calling for models that can adapt dynamically rather than relying solely on static long-term signals. Prior request-aware approaches fall into three strands: (i) request augmentation (Zhang et al., 2011; He et al., 2022), which conditions sequential models on user-generated tags or requests to capture short-term intent but relies on shallow representations; (ii) language-to-item representations, which pretrain encoders to bridge natural language and items (Hou et al., 2024) or model sequences directly in language space (Li et al., 2023), improving coverage and transfer but requiring domain-specific pretraining or fine-tuning and potentially adding inference cost; and (iii) LLM-based methods, which leverage large language models for recommendation via semantic enhancement (Liu et al., 2024), constrained generation (Liao et al., 2025), or listwise reasoning re-rankers (Zhang et al., 2025), though effective, they are often overly complex and demand high computation and latency. These limitations motivate lightweight mechanisms that condition strong sequential backbones on immediate requests.

In the request-aware sequential recommendation mentioned above, retraining or fully fine-tuning complex Transformer-based backbones is computationally prohibitive for real-time adaptation. Since these models already encode long-term preference signals, it is common to keep the backbone frozen and introduce lightweight modules for parameter-efficient adaptation (Su et al., 2025), as in natural language processing (NLP) (Son et al., 2025) and computer vision (CV) (Qin et al., 2024). Such approaches include prefix/prompt tuning (Li & Liang, 2021; Lester et al., 2021), which prepends small vectors for only coarse control; adapter modules (Houlsby et al., 2019), which insert trainable layers but increase latency; and latent token insertion (Sun et al., 2025), which offers flexible conditioning at the cost of additional parameters. Masking methods instead stand out by learning task-dependent masks over weights or activations, enabling reversible and fine-grained control without retraining (Zhao et al., 2020; Ansell et al., 2022; Litschko et al., 2022; Tao et al., 2023; Svirsky et al., 2024), though their potential for conditioning on natural-language requests remains under-

explored. Overall, prior work on parameter-efficient adaptation primarily targets task or domain transfer rather than the instant, request-driven control needed in sequential recommendation.

# 3 METHODOLOGY: CONTROL VIA REQUEST-AWARE MASKING FOR EDITING RECOMMENDERS

## 3.1 TASK DEFINITION

Let $\mathcal{U}$ and $\mathcal{I}$ denote the sets of users and items, respectively. For a user $u \in \mathcal{U}$, we represent the historical interaction sequence as $\boldsymbol{s}_u = (i_1, \ldots, i_T)$ with $i_t \in \mathcal{I}$, $t \in \{1, 2, ..., T\}$, and denote the ground-truth next item by $i^\star_{T+1} \in \mathcal{I}$. In addition to these common notations of sequential recommendation, in the request-aware scenario, at time step $T+1$, the user $u$ provides a natural-language request $\mathbf{q}_u$, which specifies the user's immediate intent.

We consider a sequential recommender $f_\theta$ with parameters $\theta$. Given the interaction sequence $\boldsymbol{s}_u$ and request $\mathbf{q}_u$ of user $u$, the model assigns a relevance score to each $i \in \mathcal{I}$ and predicts

$$\hat{i}_{T+1} = \arg\max_{i \in \mathcal{I}} f_\theta(i \mid \boldsymbol{s}_u, \mathbf{q}_u). \tag{1}$$

Ideally, we want this prediction to coincide with the ground-truth next item, i.e., $\hat{i}_{T+1} = i^\star_{T+1}$. The overall training objective is to maximize the total conditional log-likelihood of ground-truth next items over all users, i.e.,

$$\max \sum_{u \in \mathcal{U}} \log p\left(i^\star_{T+1} \mid \boldsymbol{s}_u, \mathbf{q}_u; \theta\right), \tag{2}$$

which amounts to encouraging the model to assign the highest probability to the ground-truth next item $i^\star_{T+1}$ given both the historical sequence and the accompanying request, consistent with the ideal case of Equation (1). In contrast to conventional sequential recommendation that relies solely on $\boldsymbol{s}_u$ and the backbone $f_\theta$, this formulation explicitly incorporates $\mathbf{q}_u$, allowing the model to reconcile long-term preferences with immediate intent.

To optimize Objective (2), existing methods take three main routes. Some manipulate $\boldsymbol{s}_u$, e.g., by augmenting or transforming it with $\mathbf{q}_u$ (He et al., 2022; Li et al., 2023; Hou et al., 2024; Liu et al., 2024), but such strategies often yield shallow control. Others introduce auxiliary request-aware modules that fuse with the backbone (Liao et al., 2025; Zhang et al., 2025), at the cost of added latency and complexity. A more direct option is to fine-tune or retrain the backbone parameters $\theta$ based on $\mathbf{q}_u$, but this is computationally expensive and impractical—since $\theta$ is often trained, tested and frozen in practice. Thus, the key challenge is to control the frozen backbone model given $\mathbf{q}_u$.

This motivates a mapping $\mathcal{F}_\phi$ with trainable parameters $\phi$, which transforms $\mathbf{q}_u$ into the *control signal* vector $\boldsymbol{m} \in \mathbb{R}^d$. Then, we apply $\boldsymbol{m}$ to $\theta$ through a series of operations $C_{\boldsymbol{m}}(\theta)$ to obtain the edited parameters $\theta'$. Therefore, starting from Objective (2), we can rewrite our goal as finding

$$\phi^\star = \arg\max_{\phi} \sum_{u \in \mathcal{U}} \log p\left(i^\star_{T+1} \mid \boldsymbol{s}_u, \theta'\right), \quad \text{where } \theta' = C_{\boldsymbol{m}}(\theta), \; \boldsymbol{m} = \mathcal{F}_\phi(\mathbf{q}_u). \tag{3}$$

## 3.2 VARIATIONAL MOTIVATION FOR MODEL CONTROL

Equation (3) defines the target optimization goal for request-aware sequential recommendation. Exact marginalization over all control signals is intractable, so we use variational inference (Blei et al., 2017) as a conceptual guide for designing a sparse, request-conditioned controller. Here we emphasize that this variational view serves only as theoretical motivation; the practical framework design are clarified in subsequent sections.

**Variational Lower Bound.** Because the control signals in $\boldsymbol{m}$ are actually designed to be binary (in the form of gates, details in later sections), we adopt a factorized Bernoulli prior $p(\boldsymbol{m})$ and approximate the posterior with a mean-field Bernoulli distribution $Q_\phi(\boldsymbol{m} \mid \mathbf{q}_u)$ parameterized by $\phi$ (Equation (A.4)); see Appendix A for details. Consider a single user $u \in \mathcal{U}$ in Equation (3), for

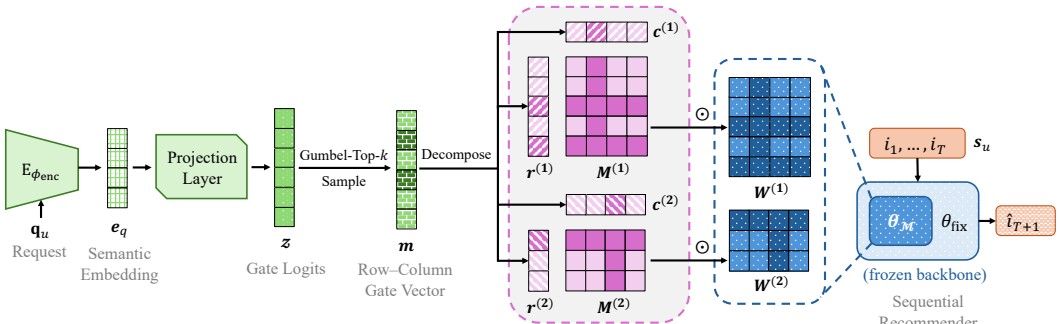

Figure 1: The overview of the proposed CRAMER framework. The figure shows the whole process of CRAMER converting a natural-language request into masks and controlling the sequential recommender (frozen backbone). The gray area with a pink dashed border represents the "Row–Column Gating Masks" paragraph in Section 3.3.

such $Q_\phi(\boldsymbol{m} \mid \mathbf{q}_u)$ and $p(\boldsymbol{m})$, the marginal likelihood admits the variational lower bound:

$$
\log p\big(i_{T+1}^\star \mid \boldsymbol{s}_u, \theta'\big) \;\geq\; \int Q_\phi(\boldsymbol{m} \mid \mathbf{q}_u) \log p\big(i_{T+1}^\star \mid \boldsymbol{s}_u, \theta'\big) \, \mathrm{d}\boldsymbol{m}
$$
$$
- \, \mathrm{KL}[Q_\phi(\boldsymbol{m} \mid \mathbf{q}_u) \,\|\, p(\boldsymbol{m})], \tag{4}
$$

with the evidence lower bound (ELBO)

$$
\mathcal{L}_{\mathrm{ELBO}}(u) = \mathbb{E}_{\boldsymbol{m} \sim Q_\phi}\Big[ \log p\big(i_{T+1}^\star \mid \boldsymbol{s}_u, \theta'\big) \Big] \;-\; \mathrm{KL}[Q_\phi(\boldsymbol{m} \mid \mathbf{q}_u) \,\|\, p(\boldsymbol{m})]. \tag{5}
$$

For the detailed derivation of Equation (5), please refer to Appendix A. In practice, however, this expectation is not explicitly optimized during training. Instead, we approximate it using Gumbel–Top-$k$ sampling with a straight-through estimator (see Sections 3.3 and 3.4), which is a more straightforward approach and better suited for recommender systems.

**Training Objective.** Motivated by Equation (5), we construct a tractable surrogate training objective with a KL term that admits a closed-form expression (Equation (A.5)) under the factorized Bernoulli assumption. We denote by $\ell(\hat{i}_{T+1}, i_{T+1}^\star)$ the predictive loss, i.e., the original training loss used by the backbone recommender. Our training objective is

$$
\mathcal{L}(\phi) \;=\; \frac{1}{|\mathcal{U}|} \sum_{u \in \mathcal{U}} \bigg[ \underbrace{\ell\big(\hat{i}_{T+1}^{(u)}, \, i_{T+1}^{\star (u)}\big)}_{\text{predictive loss}} + \lambda_{\mathrm{KL}} \cdot \underbrace{\frac{1}{d} \mathrm{KL}\big[ Q_\phi(\boldsymbol{m} \mid \mathbf{q}_u) \,\|\, p(\boldsymbol{m}) \big]}_{\text{KL regularizer}} \bigg]. \tag{6}
$$

where $d$ is the dimension of $\boldsymbol{m}$. The Objective (6) consists of two complementary terms. The first is the predictive loss, which directly drives the model to rank the ground-truth item highest given the historical sequence and request, ensuring recommendation accuracy. The second is the KL regularizer, which encourages the posterior distribution $Q_\phi$ to stay close to the sparsity prior $p(\boldsymbol{m})$, thereby enforcing compact and stable control. Note that we divide the KL term by the dimension $d$ of $\boldsymbol{m}$ to normalize its scale, preventing it from dominating as $d$ increases and ensuring a balanced trade-off between predictive accuracy and sparsity control.

Based on Objective (6), we propose Control via Request-Aware Masking for Editing Recommenders (CRAMER). At a high level, CRAMER adapts a frozen sequential recommender to natural-language requests by learning lightweight binary gate vectors that map each request to masks over a pretrained backbone. The edited model fuses long-term preferences with immediate intent, enabling rapid, no-retraining adaptation while preserving fine-grained control. Figure 1 overviews the framework and the following sections detail its components.

### 3.3 REQUEST-TO-MASK ADAPTATION

As discussed in Section 2, masking parameters of a deep model provides a lightweight yet expressive mechanism for model control. In this section, we introduce how CRAMER converts natural-language requests into masks that used to control the backbone.

**Request Embedding.** We begin by describing how CRAMER encodes a natural-language request into an embedding that conditions the recommendation model (frozen backbone) $f_\theta$. Unlike historical interaction sequences, requests are diverse and may contain negations, constraints, or attribute-specific preferences. To extract their semantics, we introduce a lightweight request encoder $\mathrm{E}_{\phi_{\mathrm{enc}}}$ based on a pretrained language model (PLM). Given a request $\mathbf{q}_u$, we tokenize it and obtain contextualized token embeddings, which are mean-pooled across all tokens to form a stable representation (Mosbach et al., 2020). Formally,

$$\boldsymbol{e}_q \;=\; \mathrm{E}_{\phi_{\mathrm{enc}}}(\mathbf{q}_u) \in \mathbb{R}^h,$$

where $h$ is the hidden dimension. The resulting semantic embedding $\boldsymbol{e}_q$ summarizes the user's immediate intent, allowing CRAMER to capture request semantics in a modular form while remaining compatible with the frozen sequential recommender.

**Defining the Controllable Subset.** For a Transformer-based sequential recommender system $f_\theta$, we identify a subset of its parameters as *controllable* subset $\theta_M$ that is crucial and suitable for being masked. In Transformer architectures, FFNs constitute the majority of parameters and act as key–value memories (Geva et al., 2020; Gerber, 2025); selectively masking them directly modulates what the model "remembers." Moreover, attention heads often exhibit redundancy (Michel et al., 2019), and the multi-head attention (MHA) output projection matrices $W_O$ aggregate head outputs into the residual stream (Hu et al., 2022), so masking $W_O$ provides a compact, high-leverage control knob. Guided by these observations, CRAMER supports three scopes for $\theta_M$: (i) FFNs-only, (ii) $W_O$-only, and (iii) FFNs + $W_O$. Formally, we write the maskable set of $L$ matrices as

$$\theta_M \;=\; \bigl\{ \boldsymbol{W}^{(l)} \in \mathbb{R}^{\alpha_l \times \beta_l} \bigr\}_{l=1}^{L}.$$

**Projection to Gate Logits.** Instead of assigning a mask to every parameter in $\theta_M$—which would incur prohibitive overhead given the scale of Transformer backbones—our scheme performs gating at the row and column levels (Svirsky et al., 2024). This structured design drastically reduces the number of trainable parameters while providing fine-grained, lightweight control over the backbone. Given the semantic embedding $\boldsymbol{e}_q$, we first maps it to gate logits via a linear projection layer:

$$\boldsymbol{z} \;=\; \boldsymbol{W}_{\mathrm{proj}}\boldsymbol{e}_q + \boldsymbol{b}_{\mathrm{proj}} \;\in\; \mathbb{R}^d,$$

where $d = \sum_{l=1}^{L} \bigl( \alpha_l + \beta_l \bigr)$ is the total number of row and column dimensions under the chosen scope, and $(\boldsymbol{W}_{\mathrm{proj}}, \boldsymbol{b}_{\mathrm{proj}})$ are trainable parameters.

**Constructing Sparse Binary Vector.** To achieve lightweight yet effective control, we constrain the binary vector to be $k$-hot. Let $\rho \in (0,1)$ be the drop ratio and retain exactly $k = \lceil (1-\rho)d \rceil$ active entries. To obtain them, we employ the Gumbel–Top-$k$ trick (Kool et al., 2019): for each coordinate $i$, we sample $g_i \sim \mathrm{Gumbel}(0)$ and form

$$\tilde{z}_i \;=\; z_i + g_i, \qquad i = 1, \ldots, d.$$

The indices of the $k$ largest $\tilde{z}_i$ form $S_k$, and the activated entries are

$$m_i \;=\; \mathbb{I}\{i \in S_k\}, \qquad \boldsymbol{m} \in \{0,1\}^d. \tag{7}$$

This binary vector $\boldsymbol{m}$ is precisely the instantiation of the control signal vector mentioned in Equation (3), and the first four paragraphs of this section together constitute a concrete realization of the mapping $\mathcal{F}_\phi$ described in Section 3.1.

**Row–Column Gating Masks.** In our CRAMER framework, $\boldsymbol{m}$ acts as a row-column gate vector that compactly specifies the activations of all maskable matrices in $\theta_M$. We decompose $\boldsymbol{m}$ into per-matrix segments to obtain, for each $l$, a row gate vector $\boldsymbol{r}^{(l)} \in \{0,1\}^{\alpha_l}$ and a column gate vector $\boldsymbol{c}^{(l)} \in \{0,1\}^{\beta_l}$, and define the entrywise mask

$$M_{ij}^{(l)} \;=\; r_i^{(l)} \cdot c_j^{(l)}, \quad 1 \le i \le \alpha_l,\, 1 \le j \le \beta_l.$$

Collecting all $\boldsymbol{M}^{(l)}$ and applying them entrywise to the corresponding $\boldsymbol{W}^{(l)} \in \theta_M$ yields the edited backbone $f_{\theta'}$, with parameters

$$\theta' \;=\; \bigl( \theta_{/M}, \{ \boldsymbol{W}^{(l)} \odot \boldsymbol{M}^{(l)} : \boldsymbol{W}^{(l)} \in \theta_M \} \bigr). \tag{8}$$

where $\theta_{/M}$ represents the parameters in $\theta$ except $\theta_M$. The operations in this paragraph are the specific form of $C_{\boldsymbol{m}}(\theta)$ mentioned in Equation (3). Thus, the semantic embedding $\boldsymbol{e}_q$ is converted into a structured set of row-column gating masks that modulate the frozen backbone with minimal overhead while retaining fine-grained control.

### 3.4 Learnable Components and Discrete Optimization

**Trainable Parameters.** Since the backbone $f_\theta$ is frozen, training updates only the request-to-mask (Section 3.3) module. Two components are learnable: (i) the projection layer $(\boldsymbol{W}_{\mathrm{proj}}, \boldsymbol{b}_{\mathrm{proj}})$ that maps the semantic embedding $\boldsymbol{e}_q$ to gate logits $\boldsymbol{z}$, and (ii) a subset $\phi_t$ of the request encoder parameters $\phi_{\mathrm{enc}}$ (initialized from a PLM). We consider three regimes for $\phi_t \subseteq \phi_{\mathrm{enc}}$: none (encoder fully frozen), last (fine-tune the last layer only), and all (end-to-end tuning). This offers a flexible way to balance adaptation capacity and efficiency across backbones and datasets.

**Straight-Through Training for Gating.** As we obtain $\boldsymbol{m}$ by discretely sampling $\boldsymbol{z}$ (see Equation (7)), this process blocks gradients from $\boldsymbol{m}$ to logits $\boldsymbol{z}$. To address this issue, we adopt a straight-through estimator (STE) (Bengio et al., 2013; Jang et al., 2016) with a temperature-controlled soft surrogate in the backward pass. Concretely, the forward pass uses hard $k$-hot gates from Gumbel–Top-$k$, while the backward pass propagates gradients through

$$
v_i \;=\; \frac{\exp(z_i/\tau)}{\sum_{j=1}^{d} \exp(z_j/\tau)}, \qquad i = 1, \ldots, d, \;\; \tau > 0,
$$

serving as a continuous relaxation of the binary $m_i$. This STE scheme enables end-to-end optimization of the trainable parameters $(\boldsymbol{W}_{\mathrm{proj}}, \boldsymbol{b}_{\mathrm{proj}}, \phi_t)$ despite the discrete sampling of $\boldsymbol{m}$.

## 4 Experiments and Evaluation

### 4.1 Experimental Setup

**Datasets.** We consider four representative datasets: (i) **ReDial** (Li et al., 2018), a conversational recommendation dataset with about 11.3K movie-recommendation dialogues, where users explicitly mention movies and annotate whether they have seen or liked them; (ii) **KuaiSAR** (Sun et al., 2023), a large-scale short-video interaction dataset from Kuaishou that captures both search and recommendation behaviors, containing about 19.6M actions and 6.9M items; (iii) **Beauty**, a subset of the Amazon Reviews (Hou et al., 2024) data, with approximately 701.5K reviews and 112.6K items, enriched in metadata, review text and timestamps; (iv) **CDs&Vinyl**, another subset from Amazon Reviews (Hou et al., 2024), containing about 4.8M reviews and 701.7K items, also possessing metadata, review text and timestamps. We preprocess the four datasets in a unified manner. Limited by the computing budget, we downsample the three larger datasets (KuaiSAR, Beauty, CDs&Vinyl). For more information on data preprocessing and statistics, see the Appendix B.1.

**Backbones.** We adopt two widely used Transformer-based sequential recommenders as frozen backbones. (i) **SASRec** (Kang & McAuley, 2018) employs unidirectional self-attention to model sequential dependencies in user interaction histories with high efficiency, and has become a standard baseline in sequential recommendation. (ii) **BERT4Rec** (Sun et al., 2019) adopts a bidirectional Transformer trained with a masked item prediction objective, enabling the model to capture both left and right contexts of a target position and to produce context-rich sequence representations. These two models represent the most established architectures for sequential recommendation and provide strong non–request-aware references for our study.

**Baselines.** On top of the frozen backbones, we further compare CRAMER with several state-of-the-art request-aware methods that incorporate natural-language requests. (i) **Query-SeqRec** (He et al., 2022) introduces a request encoder to represent the query and injects it into the backbone's sequential representation through concatenation or attention, enabling request-conditioned relevance scoring. (ii) **BLaIR** (Hou et al., 2024) encodes both request text and item metadata into a shared semantic space to compute similarity signals, which are then fused with the backbone's outputs to enhance ranking with request-aware semantics. (iii) **LLM-ESR** (Liu et al., 2024) leverages cached LLM-derived semantic embeddings and combines them with collaborative backbone embeddings via a lightweight adapter, providing notable benefits for long-tail users and items while keeping the backbone frozen. (iv) **REARANK** (Zhang et al., 2025) first generates an initial ranking using the backbone and then applies an LLM reranker that reasons over user history, the request, and candidate metadata to refine the list, combining sequential modeling with listwise reasoning. The integration details of these baselines with the frozen backbones are given in Appendix B.7.

| Method | ReDial | | | | | | KuaiSAR | | | | | |
|--------|--------|--------|--------|--------|--------|--------|--------|--------|--------|--------|--------|--------|
| | H@10 | H@20 | N@10 | N@20 | M@10 | M@20 | H@10 | H@20 | N@10 | N@20 | M@10 | M@20 |
| **SASRec** | | | | | | | | | | | | |
| \ | 0.426 | 0.573 | 0.373 | 0.410 | 0.344 | 0.354 | 0.430 | 0.601 | 0.346 | 0.389 | 0.306 | 0.318 |
| Query-SeqRec | 0.450 | 0.596 | 0.391 | 0.429 | 0.350 | 0.361 | 0.451 | 0.567 | 0.348 | 0.378 | 0.313 | 0.322 |
| BLaIR | 0.447 | 0.582 | 0.392 | 0.426 | 0.287 | 0.296 | 0.479 | 0.612 | 0.408 | 0.443 | 0.293 | 0.303 |
| LLM-ESR | 0.516 | 0.666 | 0.385 | 0.423 | 0.323 | 0.334 | 0.496 | 0.628 | 0.392 | 0.427 | 0.343 | 0.351 |
| REARANK | 0.549 | 0.684 | 0.414 | 0.449 | 0.408 | 0.417 | 0.538 | 0.645 | 0.409 | 0.437 | 0.366 | 0.374 |
| CRAMER (Ours) | **0.578*** | **0.694*** | **0.428*** | **0.456*** | **0.413** | **0.421** | **0.556*** | **0.748*** | **0.436*** | **0.484*** | **0.391*** | **0.405*** |
| **BERT4Rec** | | | | | | | | | | | | |
| \ | 0.421 | 0.542 | 0.355 | 0.387 | 0.272 | 0.281 | 0.436 | 0.591 | 0.366 | 0.407 | 0.311 | 0.322 |
| Query-SeqRec | 0.462 | 0.563 | 0.347 | 0.373 | 0.307 | 0.315 | 0.464 | 0.577 | 0.364 | 0.393 | 0.349 | 0.358 |
| BLaIR | 0.466 | 0.654 | 0.395 | 0.442 | 0.333 | 0.348 | 0.480 | 0.652 | 0.401 | 0.445 | 0.339 | 0.352 |
| LLM-ESR | 0.515 | 0.668 | 0.427 | 0.465 | 0.358 | 0.369 | 0.530 | 0.715 | 0.389 | 0.436 | 0.324 | 0.338 |
| REARANK | 0.536 | 0.680 | 0.388 | 0.424 | 0.355 | 0.366 | 0.566 | 0.691 | 0.416 | 0.448 | 0.355 | 0.364 |
| CRAMER (Ours) | **0.580*** | **0.753*** | **0.451*** | **0.497*** | **0.376*** | **0.389*** | **0.598*** | **0.717** | **0.434*** | **0.467*** | **0.382*** | **0.390*** |

| Method | Beauty | | | | | | CDs&Vinyl | | | | | |
|--------|--------|--------|--------|--------|--------|--------|--------|--------|--------|--------|--------|--------|
| | H@10 | H@20 | N@10 | N@20 | M@10 | M@20 | H@10 | H@20 | N@10 | N@20 | M@10 | M@20 |
| **SASRec** | | | | | | | | | | | | |
| \ | 0.442 | 0.574 | 0.385 | 0.419 | 0.338 | 0.348 | 0.480 | 0.658 | 0.398 | 0.444 | 0.360 | 0.374 |
| Query-SeqRec | 0.479 | 0.606 | 0.352 | 0.384 | 0.302 | 0.313 | 0.511 | 0.698 | 0.406 | 0.454 | 0.355 | 0.370 |
| BLaIR | 0.495 | 0.622 | 0.421 | 0.453 | 0.348 | 0.357 | 0.525 | 0.627 | 0.450 | 0.477 | 0.349 | 0.357 |
| LLM-ESR | 0.503 | 0.701 | 0.445 | 0.495 | 0.379 | 0.395 | 0.560 | 0.699 | 0.434 | 0.470 | 0.346 | 0.355 |
| REARANK | 0.548 | 0.681 | 0.474 | 0.509 | 0.335 | 0.344 | 0.612 | 0.719 | 0.454 | 0.481 | 0.378 | 0.385 |
| CRAMER (Ours) | **0.574*** | **0.735*** | **0.489*** | **0.531*** | **0.385** | **0.397** | **0.619** | **0.726*** | **0.472*** | **0.498*** | **0.397*** | **0.404*** |
| **BERT4Rec** | | | | | | | | | | | | |
| \ | 0.409 | 0.551 | 0.331 | 0.368 | 0.323 | 0.334 | 0.416 | 0.547 | 0.300 | 0.334 | 0.291 | 0.301 |
| Query-SeqRec | 0.434 | 0.534 | 0.357 | 0.382 | 0.334 | 0.341 | 0.459 | 0.614 | 0.330 | 0.371 | 0.283 | 0.292 |
| BLaIR | 0.459 | 0.627 | 0.364 | 0.407 | 0.315 | 0.327 | 0.462 | 0.617 | 0.412 | 0.452 | 0.333 | 0.345 |
| LLM-ESR | 0.493 | 0.594 | 0.346 | 0.373 | 0.319 | 0.327 | 0.503 | 0.692 | 0.438 | 0.487 | 0.311 | 0.325 |
| REARANK | 0.509 | 0.693 | 0.379 | 0.426 | 0.331 | 0.346 | 0.571 | 0.684 | 0.423 | 0.452 | 0.367 | 0.376 |
| CRAMER (Ours) | **0.539*** | **0.734*** | **0.396*** | **0.447*** | **0.345*** | **0.359*** | **0.583*** | **0.695** | **0.487*** | **0.516*** | **0.433*** | **0.441*** |

Table 1: Overall results of four baselines and our CRAMER. H@$k$, N@$k$, and M@$k$ denote HR@$k$, NDCG@$k$, and MRR@$k$, respectively (averaged over five runs). For each setting, the boldface refers to the highest result, and the underline indicates the second best result. "*" marks statistically significant improvements after BH procedure (FDR = 0.05) across all 48 t-tests.

**Optional PLMs.** As mentioned in Section 3.3, the request encoder $E_{\phi_{enc}}$ is initialized by a PLM based on Transformer. We consider four PLMs to cover the efficiency–accuracy spectrum: (i) **BERT style** (Devlin et al., 2019), a classic encoder. (ii) **RoBERTa style** (Liu et al., 2019), a robust medium encoder. (iii) **MiniLM style** (Wang et al., 2020), a very lightweight encoder. (iv) **ModernBERT style** (Warner et al., 2024), a modern high-capacity base encoder. All encoders use default text preprocessing, and we apply mean-pooling over the final hidden states (Section 3.3) to produce the semantic embedding $e_q$, which we find more stable than single-token pooling when handling diverse request phrasing (Mosbach et al., 2020).

**Evaluation Metrics.** We adopt widely used ranking metrics in recommender system evaluation. Specifically, we report **HR@$k$** (Hit Ratio), **NDCG@$k$** (Normalized Discounted Cumulative Gain) and **MRR** (Mean Reciprocal Rank) at cutoff values $k \in \{10, 20\}$. These are very commonly used metrics in recommender system evaluation.

## 4.2 Overall Performance

Following previous papers (Kang & McAuley, 2018; Liu et al., 2024), we randomly sample 100 items that the user has not interacted with as the negatives paired with the only ground-truth positive for calculation of the metrics. Table 1 reports the overall performance of four baselines and our proposed CRAMER on four benchmark datasets under two frozen Transformer backbones.

**Aggregate Comparison.** From the results, CRAMER consistently outperforms all baselines across datasets and metrics under both SASRec and BERT4Rec backbones. Intuitively, we observe that CRAMER achieves the best results on all metrics across all experiments. After applying Benjamini–Hochberg (BH) procedure (FDR = 0.05) across all 48 paired t-tests, CRAMER remains significantly better than the strongest baseline in 41 settings (85.42%), indicating consistent and robust improve-

ments. This demonstrates that the request-aware masking mechanism effectively augments sequential recommenders, delivering more accurate predictions without requiring full fine-tuning.

**Results by Dataset and Backbone.** Across datasets, CRAMER shows clear advantages, particularly on large-scale datasets like KuaiSAR and CDs&Vinyl, where it substantially outperforms embedding-based and generative baselines, showing its ability to integrate long-term preferences with immediate requests. On smaller datasets (e.g., ReDial), it still yields consistent gains, underscoring robustness. Across backbones, CRAMER delivers stable improvements on both SASRec and BERT4Rec by incorporating request semantics, thereby addressing the limitation that both models rely solely on users' historical interactions for prediction. Overall, CRAMER is a general and effective framework for request-aware sequential recommendation across datasets and architectures.

**Intuitive User Case Study.** To provide a more intuitive understanding of how our approach improves recommendation quality, we further conduct a case study on five specific users from the CDs&Vinyl dataset. For more details, please refer to Appendix B.6.

**Summary.** In summary, CRAMER consistently outperforms embedding-based, generative, and reasoning-driven request-aware baselines. It achieves significant improvements across nearly all datasets, metrics, and backbones. The results in Table 1 establish CRAMER as a general, flexible and robust framework that effectively integrates long-term preferences with immediate intent while remaining efficient under frozen sequential backbones.

### 4.3 SENSITIVITY ANALYSIS OF HYPERPARAMETERS

While the overall results in Section 4.2 demonstrate the effectiveness of CRAMER, it is important to understand how different hyperparameters influence performance. We therefore conduct several sensitivity studies to disentangle the contribution of each design choice. In each backbone $\times$ dataset experiment, we vary only the hyperparameter of interest while keeping all other settings fixed at their optimal values (listed in Appendix B.3). Evaluation is reported in terms of NDCG@10. Among the full set of hyperparameters we examined, two are particularly crucial: (i) drop ratio $\rho$, which determines how many units remain active under the request-to-mask mechanism; and (ii) selection of PLM used for initialize the request encoder. In this section we focus on these two factors, while additional experiments (e.g., experiments on $\theta_M$, $\lambda_{\text{KL}}$, $\phi_t$) are deferred to Appendix B.4.

**Sensitivity to Drop Ratio $\rho$.** Figure 2 reports results for $\rho \in \{0.05, 0.10, 0.15, 0.20, 0.25\}$. We find that performance is generally robust within a moderate range but deteriorates at extreme values. In most cases, a $\rho$ of around 0.10 achieves the best balance: too small a $\rho$ (i.e., almost no parameters are masked) weakens the influence of request conditioning, while too large a $\rho$ (i.e., masking too many parameters) harms the backbone's capacity. We also find that small-scale datasets tend to achieve their best performance at larger values of $\rho$, while large-scale, information-dense datasets show optimal performance at smaller $\rho$. From a theoretical analysis, we believe this is because the risk of overfitting is greater on small-scale datasets; higher sparsity imposes stronger regularization, helping to avoid overfitting and making the model focus only on the most salient request signals. On large-scale datasets, there is enough data to support more active parameters, so denser masks allow more backbone capacity to be utilized, improving expressiveness (Hoefler et al., 2021).

**Selection of PLM.** Figure 3 compares MiniLM, BERT, RoBERTa, and ModernBERT as optional PLMs, which are used to initialize the request encoders. Overall, RoBERTa and ModernBERT consistently yield the best performance: RoBERTa excels on small-scale or linguistically diverse datasets such as ReDial and Beauty, while ModernBERT dominates on large-scale or information-dense datasets such as CDs&Vinyl and KuaiSAR. In contrast, MiniLM, though computationally efficient, underperforms due to its limited capacity, and vanilla BERT trails behind its stronger successors in most cases. This demonstrates the ability of the CRAMER framework to fully leverage better and more robust language models, marking its excellence in capturing the request information.

### 4.4 EFFICIENCY AND OVERHEAD

Beyond effectiveness, another salient advantage of CRAMER is its lightweight inference behavior. Once the request-to-mask module is trained, processing a new request only requires a single forward pass with an additional lightweight projection and masking step. This design ensures that the inference cost per request remains extremely low, making the method highly suitable for

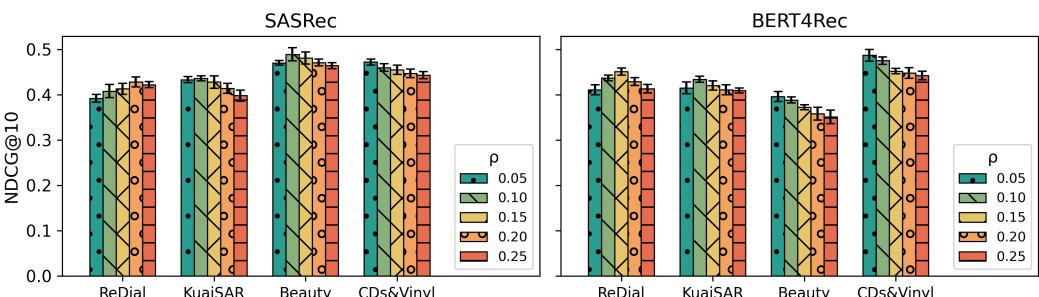

Figure 2: Sensitivity of CRAMER to drop ratio $\rho$, evaluated using NDCG@10. For each setting, five evaluations were performed, the column height shows its average result, and the error bar marks the highest and lowest results in the five evaluations.

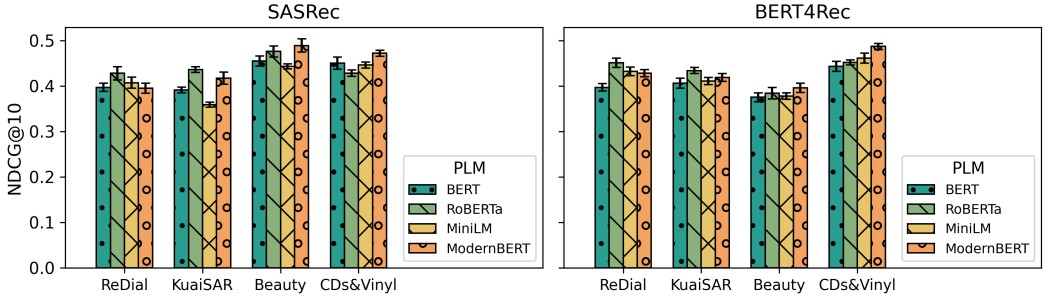

Figure 3: Impact of different PLMs on request encoding, evaluated using NDCG@10. For each setting, five evaluations were performed, the column height shows its average result, and the error bar marks the highest and lowest results in the five evaluations.

real-time recommendation scenarios. Table 2 reports the average inference cost of each method, measured as the wall-clock time and peak GPU memory usage required to process a single request under the same environment. We report the runtime and memory footprint of the vanilla SASRec and BERT4Rec backbones, as well as the incremental overhead introduced by each request-aware method. CRAMER adds only 0.018s on top of SASRec and 0.021s on top of BERT4Rec, placing it among the most efficient methods; its overhead is comparable to LLM-ESR and lower than Query-SeqRec and BLaIR, while maintaining an acceptable GPU memory footprint. In contrast, REAR-ANK, while powerful, incurs substantial overhead, requiring more than two orders of magnitude longer due to listwise reasoning across multiple candidates. These results highlight that CRAMER strikes a favorable balance between accuracy and efficiency: once trained, it consistently delivers superior recommendation quality while keeping both runtime and memory usage minimal, offering a practical and scalable solution for real-time deployment.

| Method | SASRec | | BERT4Rec | |
|---|---|---|---|---|
| | Runtime (s) | GPU Memory (MiB) | Runtime (s) | GPU Memory (MiB) |
| *Vanilla Backbone* | 0.033 | 2024.1 | 0.038 | 2119.6 |
| + Query-SeqRec | + 0.021 | + 1587.5 | + 0.023 | + 1620.4 |
| + BLaIR | + 0.029 | + 1125.4 | + 0.027 | + 1102.6 |
| + LLM-ESR | + 0.016 | + 1236.2 | + 0.018 | + 1375.3 |
| + REARANK | + 9.256 | + 9824.7 | + 9.184 | + 9412.5 |
| + CRAMER (Ours) | + 0.018 | + 1355.6 | + 0.021 | + 1408.1 |

Table 2: Inference efficiency comparison for both SASRec and BERT4Rec backbones. Runtime and GPU memory usage are measured as average per-request cost under identical settings. "Vanilla Backbone" row reports the backbone-only cost, and the subsequent rows present the additional runtime or GPU memory incurred when equipping the backbone with each request-aware method.

## 4.5 MASK INTERPRETABILITY

To evaluate whether CRAMER's request-conditioned masks reflect meaningful semantics, we conduct an interpretability study on the ReDial dataset. Using genre labels obtained via the Open Movie Database[1] (one of ReDial's original data sources), we determine whether a recommended movie belongs to the romance-related category. For a sampled group of 100 users, we issue six types of requests around the "romance" concept: (1) clear positive, (2) ambiguous positive, (3) rare-term positive, (4) clear opposite, (5) ambiguous opposite, and (6) rare-term opposite, to influence their recommendation results. For each request, we measure the proportion of romance-related items in the top-10 recommendations and compute the mean and variance across users.

| Request Type | Request Text | Before (Avg, Var) | After (Avg) | After (Var) |
|---|---|---|---|---|
| Clear Positive | *"I'd like a romantic comedy."* | | 0.432 ↑ | 0.0244 |
| Ambiguous Positive | *"Something sweet and heartwarming."* | | 0.345 ↑ | 0.0253 |
| Rare-Term Positive | *"I want an offbeat, slow-burn emotional drama."* | Avg: 0.286 | 0.312 ↑ | 0.0371 |
| Clear Opposite | *"Please avoid romantic movies."* | Var: 0.0218 | 0.135 ↓ | 0.0187 |
| Ambiguous Opposite | *"Maybe something less focused on love."* | | 0.204 ↓ | 0.0198 |
| Rare-Term Opposite | *"Skip movies with amour-driven plots."* | | 0.253 ↓ | 0.0389 |

Table 3: The proportion of romance-related movies in the top-10 recommendations before and after issuing different request types (100 users). The "Before" column shows the mean and variance without considering any requests, while the two "After" columns show the mean and variance after considering the corresponding requests.

Across all six request types, the results in Table 3 demonstrate that CRAMER produces consistent and semantically aligned shifts in the recommendation distribution. Clear positive requests substantially increase the proportion of romance-related movies in the top-10 list, while clear opposite requests consistently decrease it. This monotonic behavior supports that CRAMER's request-conditioned masks faithfully encode the intended semantics. Ambiguous requests also induce smaller but still noticeable shifts in the expected direction, indicating that CRAMER can interpret indirect user intent. Moreover, rare-term requests can still lead to appropriate adjustments. Although we can observe an increase in variance, it remains within acceptable limits, suggesting robustness to infrequent phrasing. These findings demonstrate that CRAMER provides both interpretable and stable control over the backbone model.

## 5 CONCLUSION

In this paper, we introduced CRAMER, a lightweight framework for request-aware sequential recommendation that treats natural-language requests as control inputs. We first formalized the problem and proposed a variationally motivated training objective with KL regularization and STE, enabling stable and efficient optimization. CRAMER encodes each request into a semantic embedding, which is projected into structured row–column masks that modulate frozen Transformer backbones, providing fine-grained and efficient control without retraining. Through extensive experiments on four benchmark datasets and two backbones, we demonstrated that CRAMER consistently outperforms strong request-aware baselines across multiple metrics while incurring minimal runtime and memory overhead. Overall, CRAMER establishes a new paradigm for controllable, efficient, and scalable integration of immediate user intent into sequential recommendation.

## ETHICS STATEMENT

All authors have carefully read and fully acknowledge the ICLR Code of Ethics `https://iclr.cc/public/CodeOfEthics`, and we confirm that this submission strictly complies with the principles outlined by the Code. Our study focuses on methodological contributions to request-aware sequential recommendation and does not involve human subjects, private or sensitive user

---

[1] `https://www.omdb.org`

data, or any potentially harmful applications. All datasets used in this work are publicly available benchmarks, which have been widely adopted in prior recommendation research.

Our method does not introduce risks related to fairness, discrimination, or privacy beyond those already present in the benchmark datasets. No external sponsorship or conflicts of interest influence this work. We therefore believe that our submission raises no ethical concerns.

## REPRODUCIBILITY STATEMENT

We have made significant efforts to ensure the reproducibility of our work. The implementation of CRAMER, the proposed framework in Section 3 is anonymously available at `https://anonymous.4open.science/r/CRAMER-SubmissionVer-1694/`. For the theoretical part, the complete derivation and explanation are in Appendix A. For the experimental part, the details are in Appendix B. Together, these resources enable independent researchers to reproduce our experiments and validate our claims.

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

# A   DERIVATION OF THE TRAINING OBJECTIVE

We start from the request-aware maximum-likelihood formulation in Objective (2):

$$\max \sum_{u \in \mathcal{U}} \log p\big(i^\star_{T+1} \mid \boldsymbol{s}_u, \theta'\big), \quad \text{where } \theta' = C_{\boldsymbol{m}}(\theta), \ \boldsymbol{m} = \mathcal{F}_\phi(\mathbf{q}_u).$$

According to our definition in Section 3.1, $f_\theta$ is a frozen sequential backbone with parameters $\theta$, and only the subset $\theta_M$ is subject to request-conditioned masking. Let $\boldsymbol{m} \in \{0, 1\}^d$ denote the row-column gate vector (Section 3.3) that selects row/column activations for all matrices in $\theta_M$. In Section 3.2 we adopt variational inference as a motivation for a tractable surrogate. Below we present the ELBO derivation, and then clarify how our practical training aligns with it under hard $k$-hot control. In the original Inequality (4), we use integral to characterize the abstract "control signals", but $\boldsymbol{m} \in \{0, 1\}^d$ is actually a binary vector, so in the derivation in this section, we use summation instead of integral.

**Marginal Likelihood as a Sum over Masks.** For a single user $u \in \mathcal{U}$, the conditional likelihood is marginalized over all gates in $\boldsymbol{m}$:

$$\log p\big(i^\star_{T+1} \mid \boldsymbol{s}_u, \theta'\big) = \log p\big(i^\star_{T+1} \mid \boldsymbol{s}_u, \mathbf{q}_u, \boldsymbol{m}; \theta\big)$$
$$= \log \sum_{\boldsymbol{m} \in \{0,1\}^d} p\big(i^\star_{T+1} \mid \boldsymbol{s}_u, \mathbf{q}_u, \boldsymbol{m}; \theta\big) \ p(\boldsymbol{m}). \tag{A.1}$$

**Prior Distribution.** We use a request-agnostic sparsity prior. Aligned with Section 3.3, we consider a factorized Bernoulli prior with activation rate equal to the actual keep ratio:

$$p(\boldsymbol{m}) = \prod_{i=1}^{d} \pi_0^{m_i} (1 - \pi_0)^{1-m_i}, \qquad \pi_0 := \frac{k}{d}, \tag{A.2}$$

so that the prior mean exactly matches the realized budget of $k = \lceil (1 - \rho)d \rceil$ active gates (note that $\pi_0$ may differ slightly from $1 - \rho$ due to the ceiling operation).

**Variational Lower Bound.** Introduce a request-conditioned variational distribution $Q_\phi(\boldsymbol{m} \mid \mathbf{q}_u)$, parameterized by $\phi$. Multiply and divide inside the sum by $Q_\phi$, and apply Jensen's inequality:

$$\log p\big(i^\star_{T+1} \mid \boldsymbol{s}_u, \theta'\big) = \log \sum_{\boldsymbol{m} \in \{0,1\}} Q_\phi(\boldsymbol{m} \mid \mathbf{q}_u) \frac{p\big(i^\star_{T+1} \mid \boldsymbol{s}_u, \mathbf{q}_u, \boldsymbol{m}; \theta\big) \ p(\boldsymbol{m})}{Q_\phi(\boldsymbol{m} \mid \mathbf{q}_u)}$$
$$= \log \mathbb{E}_{\boldsymbol{m} \sim Q_\phi(\cdot \mid \mathbf{q}_u)} \left[ \frac{p\big(i^\star_{T+1} \mid \boldsymbol{s}_u, \theta'\big) \ p(\boldsymbol{m})}{Q_\phi(\boldsymbol{m} \mid \mathbf{q}_u)} \right]$$
$$\geq \mathbb{E}_{\boldsymbol{m} \sim Q_\phi(\cdot \mid \mathbf{q}_u)} \big[ \log p\big(i^\star_{T+1} \mid \boldsymbol{s}_u, \theta'\big) + \log p(\boldsymbol{m}) - \log Q_\phi(\boldsymbol{m} \mid \mathbf{q}_u)] \big]$$
$$= \underbrace{\mathbb{E}_{\boldsymbol{m} \sim Q_\phi}\Big[ \log p\big(i^\star_{T+1} \mid \boldsymbol{s}_u, \theta'\big) \Big]}_{\text{prediction term}} - \underbrace{\mathbb{E}_{\boldsymbol{m} \sim Q_\phi}\Big[ \log \tfrac{Q_\phi(\boldsymbol{m} \mid \mathbf{q}_u)}{p(\boldsymbol{m})} \Big]}_{\text{КЦ}Q_\phi(\boldsymbol{m} \mid \mathbf{q}_u) \,\|\, p(\boldsymbol{m})]}.$$

The above is the complete derivation of Equation (5). Note that $p\big(i^\star_{T+1} \mid \boldsymbol{s}_u, \mathbf{q}_u, \boldsymbol{m}; \theta\big) = p\big(i^\star_{T+1} \mid \boldsymbol{s}_u, \theta'\big)$. Hence, the per-user evidence lower bound (ELBO) is

$$\mathcal{L}_{\text{ELBO}}(u) = \mathbb{E}_{\boldsymbol{m} \sim Q_\phi}\Big[ \log p\big(i^\star_{T+1} \mid \boldsymbol{s}_u, \theta'\big) \Big] - \text{KL}[Q_\phi(\boldsymbol{m} \mid \mathbf{q}_u) \,\|\, p(\boldsymbol{m})]. \tag{A.3}$$

which is consistent with Equation (5). In our actual algorithm the forward controller is hard $k$-hot (Gumbel–Top-$k$) with STE for gradients (Section 3.4); therefore we treat the ELBO as motivation and use a variationally inspired surrogate objective (detailed below) rather than maximizing Equation (A.3) strictly.

**Parametrization and Analytic KL.** The request-to-mask module (Section 3.3) produces logits $\boldsymbol{z} = \boldsymbol{W}_{\text{proj}} \boldsymbol{e}_q + \boldsymbol{b}_{\text{proj}} \in \mathbb{R}^d$ from the semantic embedding $\boldsymbol{e}_q$. We adopt a mean-field Bernoulli parameterization for regularization and monitoring:

$$Q_\phi(\boldsymbol{m} \mid \mathbf{q}_u) = \prod_{i=1}^{d} \pi_i^{m_i} (1 - \pi_i)^{1-m_i}, \qquad \pi_i = \sigma(z_i), \ \sigma(x) = \frac{1}{1+e^{-x}}. \tag{A.4}$$

With the Bernoulli prior in Equation (A.2), the KL admits a closed form:

$$\mathrm{KL}[Q_\phi \,\|\, p] = \sum_{i=1}^{d} \left[ \pi_i \log \tfrac{\pi_i}{\pi_0} + (1 - \pi_i) \log \tfrac{1-\pi_i}{1-\pi_0} \right], \qquad (A.5)$$

and we use the normalized version $\overline{\mathrm{KL}} = \frac{1}{d} \mathrm{KL}[Q_\phi \| p]$ so that the penalty does not scale with $d$.

At inference, $\boldsymbol{m}$ is instantiated as an exactly $k$-hot vector by (deterministic) Top-$k$ on $\boldsymbol{z}$ (we drop Gumbel noise). At training time, the forward path also uses hard $k$-hot masks via Gumbel–Top-$k$, while the backward path propagates gradients through a softmax surrogate with temperature (STE; Section 3.4). In parallel, we compute the Bernoulli parameters $\pi_i = \sigma(z_i)$ and apply the analytic Bernoulli–Bernoulli KL in Equation (A.5) as an auxiliary sparsity prior. We set the prior mean to the realized keep ratio, $\pi_0 = k/d$, aligning the prior with the hard budget used in the forward path. Because the forward sampler is $k$-hot while the KL assumes independent Bernoulli gates, the overall objective is not a strict ELBO; it is a variationally inspired surrogate consistent with our hard-sparsity design.

**From Likelihood to Supervised Loss.** We train with a supervised next-item loss $\ell(\cdot)$ in place of $-\log p(\cdot)$ (consistent with Section 3.2). Let $\theta' = \big( \theta_{/M}, \{ \boldsymbol{W}^{(l)} \odot \boldsymbol{M}^{(l)} : \boldsymbol{W}^{(l)} \in \theta_M \} \big)$ denote the edited backbone obtained by applying the row–column masks (Section 3.3). A per-user surrogate objective is

$$\mathcal{L}(u; \phi) = \underbrace{\ell\big( f_{\theta'}(\boldsymbol{s}_u, \mathbf{q}_u),\, i_{T+1}^{\star(u)} \big)}_{\text{predictive loss under hard } k\text{-hot forward}} + \lambda_{\mathrm{KL}} \cdot \overline{\mathrm{KL}}(\boldsymbol{z}; \pi_0), \qquad (A.6)$$

where $\overline{\mathrm{KL}}(\boldsymbol{z}; \pi_0)$ is computed from $\pi = \sigma(\boldsymbol{z})$ via Equation (A.5), and gradients through the discrete selection are enabled by STE with a temperature-controlled soft surrogate (Section 3.4). Equivalently, we can view Equation (A.6) as a single-sample Monte Carlo estimator of the predictive term (with the sample drawn by Gumbel–Top-$k$) plus an analytic KL regularizer computed on the logits.

**Final Training Objective.** Aggregating and averaging Equation (A.6) over $u \in \mathcal{U}$ yields the training objective reported in Section 3.2:

$$\mathcal{L}(\phi) = \frac{1}{|\mathcal{U}|} \sum_{u \in \mathcal{U}} \left[ \underbrace{\ell\big( \hat{i}_{T+1}^{(u)},\, i_{T+1}^{\star(u)} \big)}_{\text{predictive loss}} + \lambda_{\mathrm{KL}} \cdot \underbrace{\overline{\mathrm{KL}}(\boldsymbol{z}; \pi_0)}_{\substack{\text{KL regularizer} \\ \text{with prior mean } \pi_0}} \right].$$

The first term provides supervision for request-aware prediction under the edited backbone $f_{\theta'}$, while the second acts as an auxiliary sparsity prior that stabilizes optimization and prevents degenerate dense masks. In summary, the variational view motivates a tractable, analytically regularized surrogate that preserves strict $k$-sparsity in the forward path (via Gumbel–Top-$k$) and affords fine-grained, lightweight control over a frozen backbone.

**Discussion on Surrogate Objective.** As mentioned before, our training objective is variationally inspired rather than a strict ELBO, because the forward pass uses hard k-hot Gumbel–Top-$k$ masking while the KL term assumes independent Bernoulli gates. This type of approximation is standard in sparse gating and masking methods, where discrete control variables are optimized using continuous relaxations or STE (Bengio et al., 2013; Maddison et al., 2016; Jang et al., 2016; Louizos et al., 2017), and the KL term in such frameworks primarily functions as a sparsity-inducing regularizer rather than an exact posterior-matching term. In our setting, the mask acts as a control signal rather than a latent probabilistic variable, and empirical behavior is far more critical than variational tightness. We observe stable optimization, smooth mask activations, and reliable request-conditioned effects across datasets and backbones. Thus, although our objective is not a strict ELBO, it follows well-established practices in sparse neural control and maintains the desired inductive bias while remaining computationally tractable.

# B  DETAILS OF EXPERIMENTS

## B.1  DATA PREPROCESSING

We preprocess all four datasets (ReDial[2], KuaiSAR[3], Beauty and CDs&Vinyl[4]) in a unified manner to construct RecBole-style atomic files. For each user, we sort interactions chronologically and build the request text by leveraging and concatenating information from the three prior interactions (title, content, category, search keyword, etc.) before the current timestamp; when no prior history exists, we optionally fall back to the current interaction. To obtain positive instances, we filter interactions according to dataset characteristics: for ReDial and KuaiSAR we keep only positive interactions, while for Beauty and CDs&Vinyl we retain ratings greater than or equal to 4.0. Limited by the computing budget, we perform random downsampling on KuaiSAR, Beauty and CDs&Vinyl to use only a portion of the data in these datasets. Finally, only items and interactions that pass these filters are retained to form the atomic `.inter` and `.item` files used in training and evaluation. Table 4 shows the statistics of the preprocessed datasets.

| Dataset | #Items | #Inters | Average Chars |
|---|---|---|---|
| ReDial | 5207 | 36460 | 112.84 |
| KuaiSAR | 174895 | 260243 | 124.63 |
| Beauty | 44977 | 122485 | 226.85 |
| CDs&Vinyl | 76368 | 141213 | 973.21 |

Table 4: Statistics of the processed datasets. "#Items" represents the total number of items, "#Inters" represents the total number of interactions (each one contains a request), and "Average Chars" represents the average number of characters of requests.

## B.2  TRAINING OF BACKBONES

To control the variance, in our experiments, the parameters of both backbones (SASRec and BERT4Rec) under each dataset are trained using the default settings of the RecBole library. For specific parameter settings, please refer to RecBole v1.2.1 [5] (Zhao et al., 2021) .

## B.3  OPTIMAL SETTINGS

We summarize in Table 5 the optimal hyperparameter settings used in our experiments for each backbone $\times$ dataset configuration. All hyperparameters were tuned within predefined search ranges, and the best configuration was selected based on validation NDCG@10. Below we briefly describe each hyperparameter and its search space:

- $\rho$: the drop ratio of $m$; searched over $\{0.05, 0.10, 0.15, 0.20, 0.25\}$.
- **PLM**: the pretrained language model used as request encoder, selected from $\{$BERT, RoBERTa, MiniLM, ModernBERT$\}$.
- $\theta_M$: the part of the Transformer backbone subject to masking; chosen from $\{$FFNs, $W_O$, FFNs+$W_O\}$.
- $\phi_t$: the fine-tuning regime for the PLM; one of $\{$none (fully frozen), last (fine-tune the last layer), all (end-to-end tuning)$\}$.
- $\lambda_{\mathrm{KL}}$: the weight of the KL regularization term; tuned in $\{0.1, 0.2, 0.3, 0.4, 0.5\}$.
- **shared**: whether gates are sampled once and shared across the entire batch (1) or sampled independently per instance (0) in the training phase.
- $\tau$ (**0.7** $\rightarrow$ **0.3**): the temperature annealing schedule for Gumbel–Top-$k$ sampling; choose from $\{$linear, exponential, cosine$\}$ with a uniform start value of 0.7 and an end value of 0.3.

---

[2] https://redialdata.github.io/website/

[3] https://kuaisar.github.io/

[4] https://amazon-reviews-2023.github.io/

[5] https://recbole.io/docs/

| Backbone | Dataset | $\rho$ | PLM | $\theta_M$ | $\phi_t$ | $\lambda_{KL}$ | shared | $\tau$ (0.7 → 0.3) |
|---|---|---|---|---|---|---|---|---|
| SASRec | ReDial | 0.20 | RoBERTa | $W_O$ | last | 0.4 | 0 | cosine |
| | KuaiSAR | 0.05 | ModernBERT | FFNs+$W_O$ | all | 0.1 | 0 | cosine |
| | Beauty | 0.10 | RoBERTa | FFNs+$W_O$ | last | 0.2 | 0 | cosine |
| | CDs&Vinyl | 0.10 | ModernBERT | FFNs+$W_O$ | all | 0.1 | 0 | cosine |
| BERT4Rec | ReDial | 0.15 | RoBERTa | FFNs | last | 0.3 | 0 | cosine |
| | KuaiSAR | 0.05 | ModernBERT | FFNs+$W_O$ | all | 0.1 | 0 | cosine |
| | Beauty | 0.10 | RoBERTa | FFNs+$W_O$ | last | 0.2 | 0 | cosine |
| | CDs&Vinyl | 0.05 | ModernBERT | FFNs+$W_O$ | all | 0.2 | 0 | cosine |

Table 5: Optimal hyperparameter settings for each backbone × dataset configuration.

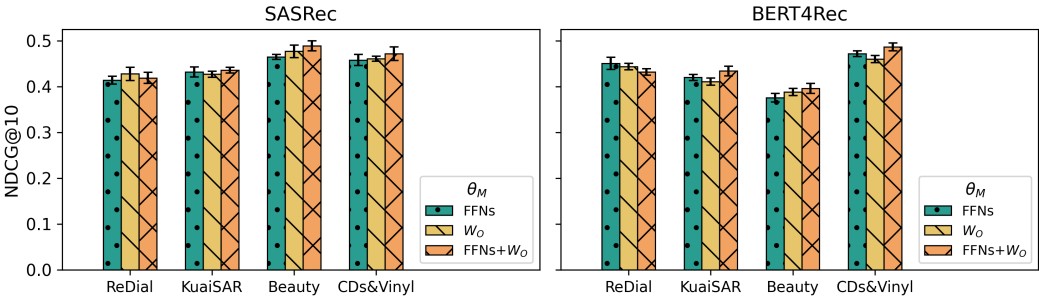

Figure 4: Impact of different scopes of $\theta_M$, evaluated using NDCG@10. For each setting, five evaluations were performed, the column height shows its average result, the the error bar marks the highest and lowest results in the five evaluations.

### B.4 DETAILED EXPERIMENTS

In this section, we present more detailed experiments. In Section 4.3, we present experiments on sensitivity to $\rho$ and PLM selection, and here we present the other experiments.

**Scopes of $\theta_M$.** We further examine the impact of different masking scopes $\theta_M$ on model performance. As described in Section 3.3, we consider three options: masking only the feed-forward networks (FFNs), masking only the output projection of multi-head attention ($W_O$), and masking both jointly (FFNs+$W_O$). The results in Figure 4 show several consistent patterns. First, the overall performance across different scopes remains relatively stable, suggesting that CRAMER is robust to the precise choice of $\theta_M$. Second, the joint scope (FFNs+$W_O$) is most often optimal, particularly on larger datasets, where combining the two sources of control enables more expressive adaptation. Third, on smaller datasets, restricting the scope to either FFNs or $W_O$ alone can be advantageous, likely because a more constrained control space reduces the risk of overfitting when training data are limited (Bejani & Ghatee, 2021). This observation aligns with the intuition that FFNs mainly act as memory slots while $W_O$ governs attention aggregation–in low-data regimes, focusing on a single component may provide more stable and interpretable modulation, while in large-scale scenarios, the joint scope is more beneficial as abundant data supports richer request-conditioned adaptations and fully exploits the complementary roles of FFNs and $W_O$. In practice, a simple principle emerges: for large-scale, information-dense datasets, applying joint masking to both FFNs and $W_O$ is generally the best choice, whereas for smaller datasets, selecting either FFNs or $W_O$ individually may be preferable. These results confirm our design motivation in Section 3.3, where both FFNs and $W_O$ were identified as high-leverage control targets for request-aware adaptation.

**Regimes of $\phi_t$.** We further investigate the effect of different fine-tuning regimes for the request encoder parameters $\phi_t$, comparing three settings: (**none**) fully frozen PLM, (**last**) tuning only the last layer, and (**all**) end-to-end tuning. The results in Figure 5 show that CRAMER is generally robust across regimes, but some clear patterns emerge. Tuning only the last layer tends to yield stable gains over the frozen setting, particularly in smaller datasets or scenarios with limited request information, where modest adaptation is sufficient and helps avoid overfitting. In contrast, end-to-end fine-tuning becomes more beneficial in large-scale or information-rich datasets, where abundant data and longer request texts can support deeper adaptation of the PLM. However, full fine-tuning may occasionally harm performance in low-data regimes, reflecting optimization instability and overfitting risks. In

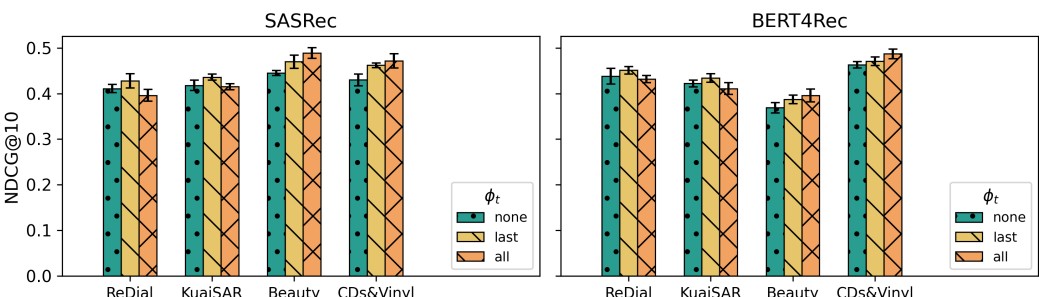

Figure 5: Impact of different regimes of $\phi_t$, evaluated using NDCG@10. For each setting, five evaluations were performed, the column height shows its average result, and the error bar marks the highest and lowest results in the five evaluations.

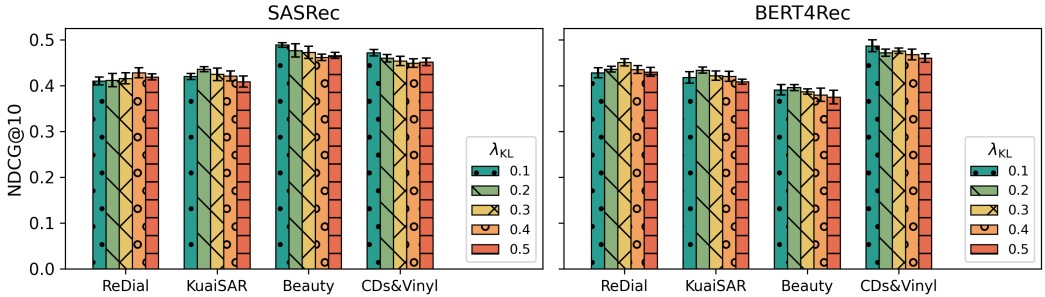

Figure 6: Sensitivity of CRAMER to weight $\lambda_{\mathrm{KL}}$, evaluated using NDCG@10. For each setting, five evaluations were performed, the column height shows its average result, and the error bar marks the highest and lowest results in the five evaluations.

practice, last-layer tuning offers a comparatively robust and reliable default, while full fine-tuning is best reserved for settings with sufficient scale and linguistic richness to fully exploit the request encoder's capacity.

**Sensitivity to the Weight of KL Regularizer $\lambda_{\mathrm{KL}}$.** We further study the influence of the KL regularization weight $\lambda_{\mathrm{KL}}$ on model performance. Figure 6 reports NDCG@10 across different values of $\lambda_{\mathrm{KL}} \in \{0.1, 0.2, 0.3, 0.4, 0.5\}$. Overall, the results indicate that CRAMER is relatively robust to the precise choice of $\lambda_{\mathrm{KL}}$, with only moderate fluctuations across datasets and backbones. On smaller datasets such as ReDial, slightly larger values (around 0.3–0.4) tend to be more effective, likely because stronger regularization prevents overfitting under limited training signals. On the contrary, on relatively larger datasets such as KuaiSAR and CDs&Vinyl, weaker regularization (around 0.1-0.2) performs best, while overly large $\lambda_{\mathrm{KL}}$ values consistently degrade performance by constraining the masks too strongly. Beauty shows an intermediate trend, where moderate values (around 0.2–0.3) strike a reasonable balance. These observations suggest a simple guideline: a small $\lambda_{\mathrm{KL}}$ is generally sufficient for large-scale datasets, while moderate values are preferable for low-data regimes. Extreme settings should be avoided, as they either under-regularize or over-constrain the request-to-mask distribution.

**Masks Shared or Not.** We study whether the gating masks are sampled once and shared across the whole mini-batch (shared=1) or sampled independently for each instance (shared=0) during training. As shown in Figure 7, the non-shared regime dominates across all datasets and both backbones: its NDCG@10 is consistently and substantially higher. Notably, the best shared result never exceeds—and often trails well behind—the non-shared results. A plausible explanation is that sharing masks across an entire batch reduces the diversity of request-conditioned adaptation signals seen during training. This compromises the model's ability to align masks closely with individual requests, leading to systematically weaker representations. By contrast, sampling masks independently per instance maintains alignment between each request and its induced control signal, enabling more

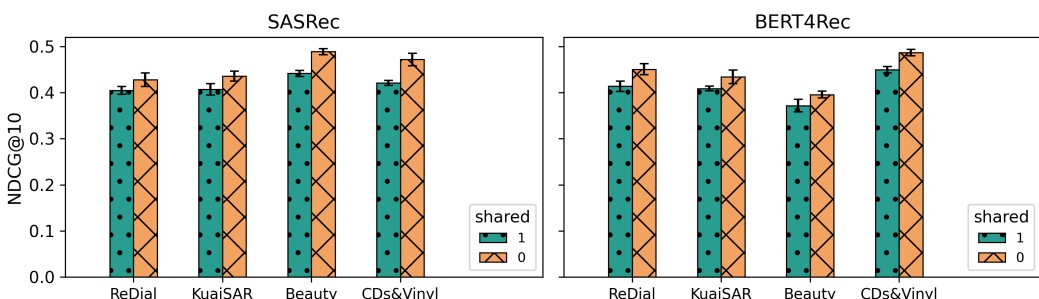

Figure 7: The effect of whether gates are sampled once and shared across the entire batch (1) or sampled independently per instance (0) in the training phase, evaluated using NDCG@10. For each setting, five evaluations were performed, the column height shows its average result, and the error bar marks the highest and lowest results in the five evaluations.

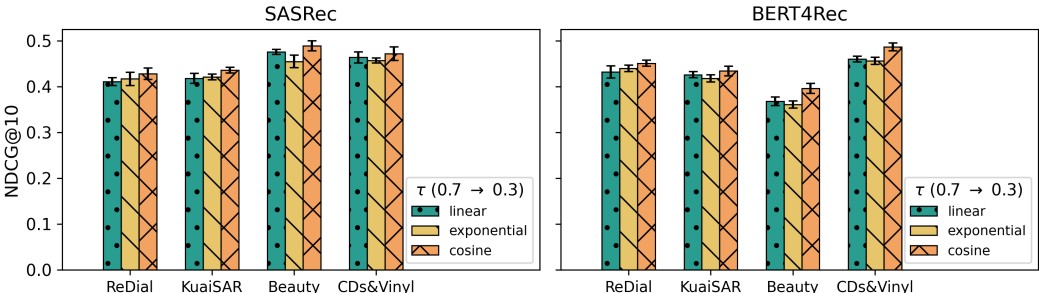

Figure 8: The effect of different temperature annealing schedule, evaluated using NDCG@10. For each setting, five evaluations were performed, the column height shows its average result, and the error bar marks the highest and lowest results in the five evaluations.

faithful request-to-mask adaptation and stronger predictive performance. From a training perspective, while the shared strategy can slightly reduce runtime overhead by avoiding per-instance sampling, this computational saving is outweighed by the clear and consistent performance degradation. Therefore, non-shared sampling should be regarded as the default choice, as it yields both more accurate and more reliable models in practice.

**Temperature Annealing Schedule.** We further compare different schedules for annealing the Gumbel–Softmax temperature $\tau$ from 0.7 to 0.3 during training, including linear, exponential, and cosine decays. Figure 8 presents the results. Overall, cosine annealing achieves the best performance in most datasets and backbones, consistently outperforming linear decay and often surpassing exponential decay by a clear margin. Linear decay provides competitive results and is generally more stable than exponential, which tends to underperform due to overly rapid decreases in temperature at the early stages of training. The superiority of cosine annealing is likely because it offers a smoother and more gradual reduction, balancing exploration and exploitation more effectively while preserving sufficient stochasticity in the mask sampling process. In practice, cosine decay can be recommended as the default schedule, while linear decay remains a reasonable alternative when simplicity is preferred. Exponential decay is less favorable, as its aggressive early cooling can lead to suboptimal convergence and weaker final accuracy.

## B.5    FURTHER DISCUSSION ON PLM

To further examine whether CRAMER is inherently unstable with respect to the choice of request encoder, we conduct an additional study using four BERT-family PLMs with increasing capacity (Tiny, Mini, Medium, and Base) as the initialization of the request encoder (the "Base" version cor-

responds to the model used in the main experiments). As shown in Table 6, we evaluate CRAMER on SASRec across four datasets using NDCG@10.

All four datasets exhibit the exact same strictly monotonic ranking among the PLM versions (Tiny < Mini < Medium < Base), which aligns precisely with their expected capacity ordering. For a single dataset, the probability of observing such a perfect ordering purely by chance is $1/24 \approx 0.0417 < 0.05$. Observing this pattern independently and consistently across all four datasets makes it highly unlikely that the variation originates from instability in CRAMER. Instead, these results indicate that CRAMER reliably leverages the semantic information provided by the request encoder: as PLM capacity improves, the encoder yields correspondingly richer representations of user requests, leading to more accurate and fine-grained control signals.

Importantly, this behavior does not represent a limitation of our framework, but rather reflects its inherently *forward-compatible* design. Even with classic and widely deployed PLMs such as BERT-base, CRAMER already achieves strong performance, and ongoing trends toward more capable yet efficient PLMs suggest that future lightweight encoders will only further enhance the model's ability to interpret user requests. Overall, the observed sensitivity arises primarily from intrinsic differences in PLM semantic expressiveness, and CRAMER remains both robust in practice and naturally aligned with continued advancements in text encoder architectures.

| BERT Version | Tiny | Mini | Medium | Base |
|---|---|---|---|---|
| ReDial | 0.403 | 0.411 | 0.419 | 0.428 |
| KuaiSAR | 0.420 | 0.425 | 0.432 | 0.436 |
| Beauty | 0.465 | 0.474 | 0.481 | 0.489 |
| CDs&Vinyl | 0.449 | 0.454 | 0.460 | 0.472 |

Table 6: Average NDCG@10 performance (five evaluations) of CRAMER under SASRec when initialized with four BERT versions of different capacity. Across all datasets, performance exhibits a strictly monotonic improvement that aligns with PLM capacity, illustrating that CRAMER faithfully leverages the semantic quality of the request encoder.

## B.6 Intuitive User Case Study

To complement the aggregate metrics, we further present an intuitive case study on five specific users from the CDs&Vinyl dataset. Table 7 lists the detailed information of these users. For every user, we compute the rank position of the ground-truth positive item among all candidates under different backbones and request-aware methods. The results are summarized in Table 8.

As shown in Table 8, vanilla SASRec and BERT4Rec often place the ground-truth item at relatively low positions, indicating their limited ability to capture the user's immediate intent. Adding request-aware baselines consistently improves the ranking quality but still exhibiting instability and fluctuations across different users. In contrast, CRAMER achieves the highest ranks for all selected users under both backbones, demonstrating more reliable alignment with the user's natural-language request. This case study provides an intuitive, per-user confirmation that CRAMER delivers consistent improvements at the individual level beyond aggregate metrics.

## B.7 Details of Combination of Baselines and Backbones

In this section, we describe how each baseline is combined with the sequential recommendation backbones (SASRec and BERT4Rec) in our experiments. The combination strategies are detailed as follows:

**Query-SeqRec.** Query-SeqRec (He et al., 2022) integrates the request encoder with the sequential backbone. The backbone models the user's historical sequence, while the request encoder provides a semantic representation of the request. These two signals are fused through concatenation, and the fused representation is used to score candidate items.

**BLaIR.** BLaIR (Hou et al., 2024) encodes item metadata and natural-language requests into a unified embedding space, such that their representations can be directly compared. The cosine similarity between the semantic embedding and the item embedding is first computed in this shared space. Specifically, the semantic similarity score between the semantic embedding $v_q$ and the item embed-

| Index | User ID | #Inters | Last Item ID |
|---|---|---|---|
| #1 | AE25K5V5RESPJ4WKCALB3ZVYYQPQ | 11 | B000008KJ8 |
| #2 | AFE66HHU55NJMALT34HEODVGEPQA | 6 | B00KNTDO3I |
| #3 | AG2CJZJORAG7SG32SYNTNHICMGOQ | 8 | B07RF4JVGJ |
| #4 | AGUPFBZ756HTU4YIF4QKQEX3NS2Q | 13 | B00SFXFCWA |
| #5 | AHQF2VXWQPUBKYR3RMJ6VDFDYUSQ | 9 | B08L47S144 |

Table 7: Detailed interaction information for the five users selected from the CDs&Vinyl dataset. For each user with a "User ID", "#Inters" represents the total number of this user's interactions, and "Last Item ID" represents the ID of the item in this user's last interaction. Because we use leave-one-out (LOO) evaluation, the item in the last interaction is the ground-truth item in evaluation.

| Method | #1 | #2 | #3 | #4 | #5 |
|---|---|---|---|---|---|
| **SASRec** | | | | | |
| \ | 18.2 | 28.4 | 27.0 | 19.0 | 17.8 |
| Query-SeqRec | 16.4 | 23.2 | 18.6 | 23.2 | 15.0 |
| BLaIR | 12.2 | 20.4 | 10.4 | 13.4 | 13.2 |
| LLM-ESR | 14.0 | 17.0 | 15.8 | 15.2 | 9.4 |
| REARANK | 13.6 | 23.2 | 11.2 | 10.0 | 11.4 |
| CRAMER (Ours) | **6.6** | **14.2** | **4.2** | **8.8** | **7.4** |
| **BERT4Rec** | | | | | |
| \ | 27.6 | 18.6 | 32.4 | 18.8 | 25.2 |
| Query-SeqRec | 21.2 | 17.2 | 14.0 | 17.0 | 21.2 |
| BLaIR | 19.4 | 11.4 | 16.2 | 13.0 | 14.2 |
| LLM-ESR | 11.4 | 13.8 | 18.2 | 7.6 | 13.8 |
| REARANK | 15.0 | 14.2 | 25.4 | 8.8 | 17.0 |
| CRAMER (Ours) | **7.0** | **8.2** | **11.4** | **5.2** | **12.0** |

Table 8: Average ranking of true positive items of users selected from CDs&Vinyl (smaller is better). For each setting, five evaluations were performed and the boldface refers to the highest ranking under the same backbone.

ding $\boldsymbol{v}_i$ is first computed by the BLaIR encoder. This score is then integrated with the collaborative score from the backbone:

$$\text{score}(i) = \gamma \cdot \cos(\boldsymbol{v}_q, \boldsymbol{v}_i) + (1 - \gamma) \cdot f_\theta(\boldsymbol{s}_u, i),$$

where $\gamma$ is a tunable fusion weight. In actual experiments, we set $\gamma$ to the optimal value on each backbonee $\times$ dataset.

**LLM-ESR.** LLM-ESR (Liu et al., 2024) augments the backbone with semantic embeddings derived from large language models. Specifically, each item $i$ is associated with both a semantic embedding $\boldsymbol{e}_i^{\text{sem}}$ (pre-computed by an LLM and projected via an adapter) and a collaborative embedding $\boldsymbol{e}_i^{\text{col}}$ (from the backbone). The user representation is similarly decomposed into $(\boldsymbol{u}^{\text{sem}}, \boldsymbol{u}^{\text{col}})$. The final score is given by:

$$\text{score}(i) = [\boldsymbol{e}_i^{\text{sem}} : \boldsymbol{e}_i^{\text{col}}]^\top [\boldsymbol{u}^{\text{sem}} : \boldsymbol{u}^{\text{col}}].$$

**REARANK.** REARANK (Zhang et al., 2025) is used as a reranking stage on top of the backbone. The backbone first generates an initial ranking of candidate items based on the user's history. These candidates, together with the request, are then passed to the LLM reranker, which performs reasoning over the top candidates and outputs a refined ranking.

## C  LIMITATIONS

Although CRAMER demonstrates strong empirical performance and practical advantages, several limitations remain.

- First, CRAMER inherits general limitations of Transformer-based sequential recommenders, including difficulty in scaling to extremely large item inventories and in generalizing to entirely new or unseen items. Future work can still draw on CRAMER's model control ideas to explore alternative paradigms (e.g., knowledge-based or generative recommendation) to mitigate some of these structural constraints.

- Second, while CRAMER handles many ambiguous or rare-term requests well in practice, overly abstract, underspecified, or even contradictory inputs may still affect performance. In future work, one possible direction is to introduce a preprocessing step, for example using an LLM to clarify or summarize user intent before applying CRAMER, which could further enhance robustness in real-world scenarios.

## D    USE OF LLMs

During the writing of this paper, LLMs were only used to aid writing by proofreading grammar and spelling, without contributing to the methodology, experiments, or results.

