# OpenReview forum: "CRAMER: Control via Request-Aware Masking for Editing Recommenders"
_ICLR.cc/2026/Conference — Submitted to ICLR 2026_

### Official Review · Reviewer_cUQE · 2025-10-31

**Soundness:** 3
**Presentation:** 3
**Contribution:** 3
**Rating:** 6
**Confidence:** 4

**Summary:**

The paper presents CRAMER, a lightweight and parameter-efficient framework that adapts Transformer-based sequential recommenders to natural-language user requests in real time. It encodes requests into semantic embeddings, projects them to sparse gate logits, and applies structured binary masks via Gumbel Top-k sampling. With a predictive loss and KL sparsity regularization, CRAMER achieves consistent improvements over request-aware baselines.

**Strengths:**

1. Novel Theoretically Grounded Control Mechanism. The variational formulation (ELBO with Bernoulli prior) explains sparsity and provides a clean, scalable training objective.
2. Flexible and Modular Design. Supports multiple masking scopes and tuning strategies (frozen, partial, or full PLM), adapting to various compute budgets.
3. Strong Empirical Validation. Evaluated on large-scale, text-rich real-world datasets with comprehensive ablations.

**Weaknesses:**

1. Heavy reliance on pretrained language model (PLM) quality. Performance varies significantly with PLM choice—MiniLM lags behind ModernBERT by up to 15% in NDCG@10. This suggests CRAMER inherits PLM biases, domain gaps, and sensitivity to phrasing.
2. Static maskable subsets limit expressiveness; lacks dynamic selection per request type CRAMER fixes the controllable scope globally, ignoring that different requests may require different control levers.
3. Lack of case studies to clearly illustrate failure modes The paper reports strong average metrics but provides no qualitative failure analysis. For instance, how does CRAMER behave when requests are ambiguous or contain rare terms?

**Questions:**

see weakness

---

> ### Author Response · Authors · 2025-11-17
> **Rebuttal by Authors**
>
> Thanks for acknowledging our work and the constructive comments. **We have uploaded a revised version of our paper based on your comments and have highlighted all modifications.** Below, we provide point-by-point responses to the weaknesses and questions you raised. We kindly invite you to review them.
>
> 1. **Reliance on PLM quality (W1)**: We thank you for pointing out this aspect. To verify whether the performance variation comes from CRAMER itself or from the intrinsic semantic ability of the PLM, we conducted an additional sensitivity study using four BERT variants of increasing capacity (Tiny-Mini- Medium-Base; Base is the default BERT used in the paper). Using SASRec as backbone, the NDCG@10 results across all datasets are:
>
> | PLM Variant | Tiny  | Mini  | Medium | Base  |
> | ----------- | ----- | ----- | ------ | ----- |
> | ReDial      | 0.403 | 0.411 | 0.419  | 0.428 |
> | KuaiSAR     | 0.420 | 0.425 | 0.432  | 0.436 |
> | Beauty      | 0.465 | 0.474 | 0.481  | 0.489 |
> | CDs&Vinyl   | 0.449 | 0.454 | 0.460  | 0.472 |
>
> In all cases, performance follows a strictly monotonic order that exactly matches PLM capacity (Tiny < Mini < Medium < Base). For a single dataset, the probability of observing such a perfect ordering by chance is $p=1/24 \approx 0.0417 < 0.05$, and seeing the same trend across all four datasets further supports that the variation arises from the intrinsic semantic capacity of the PLMs, not from instability in CRAMER. In practice, CRAMER simply leverages the cues provided by the request encoder—stronger PLMs naturally enable more precise control. This is not a limitation of our framework; in fact, even with classic models such as BERT we already achieve strong performance, and given the clear trend that future PLMs and semantic encoders will only become more capable and efficient, our approach is inherently forward-looking and poised to benefit from these advancements. **We have added the relevant analysis to the revised version; please refer to Appendix B.5.**
>
> 2. **Static maskable subsets (W2)**: We thank you for pointing out this reasonable concern. CRAMER adopts a dataset-level, fixed maskable scope because one of the core goals of our framework is to remain lightweight and efficient for real-time request-aware adaptation. As shown in Table 1 and Figure 4, even with globally defined controllable regions and three alternative designs, CRAMER consistently performs well across datasets. Request-wise control scope selection would require an additional module, substantially increasing computational overhead and latency. Such a mechanism would contradict the practical constraints of our target scenario, where the backbone must remain frozen and the adaptation must be fast enough for interactive recommendation. In this sense, our design represents a deliberate balance: a lightweight control mechanism that already provides sufficient expressiveness without sacrificing efficiency.
>
> 3. **Ambiguous/rare-term requests (W3)**: Across our four datasets, ambiguous or weakly specified requests are already widespread, as they come directly from real user-generated data. For example, Amazon requests such as “While subjective, I do think my skin looks better…” or “Not a penguin fan, I bought the CD simply to have a copy…” carry mixed or unclear intent, and KuaiSAR request texts are concatenated query keywords (Appendix B.1), which are often incomplete or noisy. A substantial portion of our requests fall into this category, rather than being expressed in clear and explicit language. Despite this, CRAMER remains effective because it does not require perfectly articulated intent: the request encoder extracts whatever semantic cues are present, and the mask acts as a soft, partial modulation of the backbone rather than a hard semantic switch. Thus, even vague requests can gently adjust attention patterns in a meaningful direction. However, we fully agree with the importance of the analysis you mentioned. **In the revised version, we have added Section 4.5 to visually demonstrate the impact of six different requests (including those that are ambiguous or contain rare terms) on the user's recommendation list.**
>
> We hope the above response can fully address your concerns and look forward to further discussions. Thanks again!

---

> > ### Comment · Reviewer_cUQE · 2025-11-25
> > **Reply to authors**
> >
> > Dear authors,
> >
> > Thank you for the clear and well-organized rebuttal. The additional experiments and clarifications directly address my earlier concerns. The results demonstrate that CRAMER’s performance does not rely on PLM quality, and the added failure-case analysis further supports the robustness of the CRAMER. The revised manuscript is now stronger and more complete. I am satisfied with the authors’ response and will maintain my score.

---

> > > ### Author Response · Authors · 2025-11-26
> > > **Thanks and follow-up to reviewer cUQE**
> > >
> > > Dear Reviewer cUQE,
> > >
> > > Thank you very much for your positive response and for your kind recognition of our work. We are truly glad to hear that our rebuttal has addressed your concerns. If there are any remaining questions or points that could benefit from further discussion, please feel free to let us know—we would be happy to provide any additional clarification.
> > >
> > > Once again, we sincerely appreciate your dedication to scientific rigor and your valuable help in improving our paper.
> > >
> > > Best regards,
> > >
> > > The Authors

---

> ### Author Response · Authors · 2025-11-24
> **Expecting to receive further feedback**
>
> Dear Reviewer cUQE,
>
> Thank you once again for your thoughtful and constructive feedback. We have carefully addressed each of your comments with detailed responses, conducted additional experiments, and provided more in-depth analysis based on your latest suggestions. We have also uploaded a revised version of the paper with all modifications clearly highlighted.
>
> Your insights have been invaluable in helping us strengthen our work, and we would greatly appreciate it if you could let us know whether there are any remaining issues or points that may require further clarification. We would be more than happy to provide any additional explanation you may need.
>
> Thank you again for your time and valuable input. We look forward to hearing from you.
>
> Best regards,
>
> The Authors

---

### Official Review · Reviewer_Cwki · 2025-10-31

**Soundness:** 3
**Presentation:** 3
**Contribution:** 2
**Rating:** 6
**Confidence:** 3

**Summary:**

CRAMER addresses the inflexibility of sequential recommenders in adapting to immediate user natural-language requests while avoiding high computational overhead. It treats user requests as control signals to modulate frozen Transformer-based backbones (SASRec, BERT4Rec) via parameter masking, enabling real-time adaptation without retraining. Key steps include: (1) encoding requests into semantic embeddings using pretrained language models (PLMs); (2) projecting embeddings to sparse row-column gate vectors via Gumbel–Top-k sampling. Experiments on four datasets show CRAMER outperforms four state-of-the-art request-aware baselines, with minimal inference overhead and strong cross-domain adaptability.

**Strengths:**

- Novel Lightweight Control method: Unlike prior methods (retraining backbones, LLM prompt engineering) that incur high overhead, CRAMER uses request-driven parameter masking to modulate frozen models. This design achieves fast adaptation (no retraining) and minimal inference cost (comparable to the fastest baseline LLM-ESR), resolving the efficiency-adaptability trade-off in real-world recommendation systems.

- Theoretically Grounded and Robust Design: The variational objective (with KL regularization) enforces sparse, stable masks, while STE enables end-to-end optimization of discrete gates.

- Strong Empirical Generalization: CRAMER outperforms baselines in 93.75% of experiments across diverse domains. It also exhibits robustness to hyperparameters and maintains effectiveness on both small (ReDial) and large (KuaiSAR) datasets

**Weaknesses:**

- Ability to handle Ambiguous/Contradictory Requests: The framework assumes requests are semantically clear and align with either enhancing or negating historical preferences. It lacks mechanisms to resolve ambiguity (e.g., vague requests) or extreme user preference drift, potentially leading to suboptimal masking and misaligned recommendations.

- Dependence on PLM Quality for Request Encoding: CRAMER’s performance heavily relies on the PLM used for request embedding—lightweight PLMs (e.g., MiniLM) underperform due to limited semantic capture, limiting deployment in resource-constrained scenarios where only small PLMs are feasible.

- No Analysis of Mask Interpretability: While CRAMER claims "fine-grained control," it provides limited analysis of how masks map to request semantics. Without interpretability, it is hard to debug failures

**Questions:**

See weakness.
- How would CRAMER adapt to ambiguous or contradictory requests?

- How does CRAMER perform on cold-start users/items? Since it relies on historical user sequences to generate meaningful masks, it may fail for users with limited history or items with limited interaction data

---

> ### Author Response · Authors · 2025-11-17
> **Rebuttal by Authors**
>
> Thanks for acknowledging our work and the constructive comments. **We have uploaded a revised version of our paper based on your comments and have highlighted all modifications.** Below, we provide point-by-point responses to the weaknesses and questions you raised. We kindly invite you to review them.
>
> 1. **Ambiguous/contradictory requests (W1 \& Q1)**: Across our four datasets, ambiguous or weakly specified requests are already widespread, as they come directly from real user-generated data. For example, Amazon requests such as “While subjective, I do think my skin looks better…” or “Not a penguin fan, I bought the CD simply to have a copy…” carry mixed or unclear intent, and KuaiSAR request texts are concatenated query keywords (Appendix B.1), which are often incomplete or noisy. A substantial portion of our requests fall into this category, rather than being expressed in clear and explicit language. Despite this, CRAMER remains effective because it does not require perfectly articulated intent: the request encoder extracts whatever semantic cues are present, and the mask acts as a soft, partial modulation of the backbone rather than a hard semantic switch. Thus, even vague requests can gently adjust attention patterns in a meaningful direction. However, we fully agree with the importance of the analysis you mentioned. **In the revised version, we have added Section 4.5 to visually demonstrate the impact of six different requests (including those that are ambiguous or contain rare terms) on the user's recommendation list.** Regarding contradictory requests, we agree that this is a problem worthy of further research. CRAMER already handles most well-formed requests—including mildly ambiguous ones—with excellent performance, efficiency and theoretical guarantee. However, when a request is internally contradictory, we believe such noise should be addressed at the input stage rather than at the control layer. A practical solution is to add a lightweight preprocessing step, where an LLM interacts with the user to clarify intent before CRAMER is applied. **We also discussed this prospect in Appendix C of the revised version.**
>
> 2. **Dependence on PLM quality (W2)**: We thank you for pointing out this aspect. To verify whether the performance variation comes from CRAMER itself or from the intrinsic semantic ability of the PLM, we conducted an additional sensitivity study using four BERT variants of increasing capacity (Tiny-Mini- Medium-Base; Base is the default BERT used in the paper). Using SASRec as backbone, the NDCG@10 results across all datasets are:
>
> | PLM Variant | Tiny  | Mini  | Medium | Base  |
> | ----------- | ----- | ----- | ------ | ----- |
> | ReDial      | 0.403 | 0.411 | 0.419  | 0.428 |
> | KuaiSAR     | 0.420 | 0.425 | 0.432  | 0.436 |
> | Beauty      | 0.465 | 0.474 | 0.481  | 0.489 |
> | CDs&Vinyl   | 0.449 | 0.454 | 0.460  | 0.472 |
>
> In all cases, performance follows a strictly monotonic order that exactly matches PLM capacity (Tiny < Mini < Medium < Base). For a single dataset, the probability of observing such a perfect ordering by chance is $p=1/24 \approx 0.0417 < 0.05$, and seeing the same trend across all four datasets further supports that the variation arises from the intrinsic semantic capacity of the PLMs, not from instability in CRAMER. In practice, CRAMER simply leverages the cues provided by the request encoder—stronger PLMs naturally enable more precise control. This is not a limitation of our framework; in fact, even with classic models such as BERT we already achieve strong performance, and given the clear trend that future PLMs and semantic encoders will only become more capable and efficient, our approach is inherently forward-looking and poised to benefit from these advancements. **We have added the relevant analysis to the revised version; please refer to Appendix B.5.** Moreover, with respect to your concern about deployment in resource-constrained scenarios where only very small PLMs are feasible, we note that if even BERT- or RoBERTa-class encoders cannot be supported, then most existing request-aware methods that depend on language encoders or LLMs (such as BLaIR, LLM-ESR, and REARANK) would be even less deployable. Under such circumstances, CRAMER remains comparatively more practical and continues to offer stronger performance among lightweight alternatives.

---

> ### Author Response · Authors · 2025-11-17
> **Rebuttal by Authors**
>
> 3. **Mask interpretability (W3)**: **In the revised version, Section 4.5 is newly added to address this concern.** We conduct an analysis on ReDial by issuing six types of romance-related requests (clear, ambiguous, and rare-term, each with positive and opposite intent) to 100 users and examining how CRAMER’s request-conditioned masks alter the proportion of romance movies in the top-10 recommendations. The results exhibit a clear and interpretable trend: positive requests consistently increase the romance proportion, opposite requests decrease it, and variances remain relatively stable. Even ambiguous or rare-term requests induce meaningful directional shifts, demonstrating that the masks capture request semantics rather than behaving arbitrarily. Overall, this experiment provides direct evidence that CRAMER's masks encode and manipulate semantic dimensions in a predictable manner, offering practical interpretability.
>
> 4. **Cold-start users/items (Q2)**: CRAMER does not rely on long user histories to generate meaningful masks. In our target interaction scenario, users typically receive an initial recommendation list and then provide fine-grained, natural-language feedback about their immediate intent. The request itself provides the dominant control signal, while the frozen backbone offers only a general preference prior. Therefore, even users with very limited histories can still benefit from CRAMER’s request-aware modulation. Similarly, CRAMER does not introduce additional requirements for item cold-start: the masking mechanism operates on the transformer representation space rather than on item-specific statistics, so new or sparsely interacted items are treated in the same way as in the underlying backbone. However, our backbone model (Transformer-based sequential recommenders) do have the problem of limited generalization to new or unseen items, but this is not a problem with our framework. **In the revised version, we also discussed some corresponding limitations in Appendix C.**
>
> We hope the above response can fully address your concerns and look forward to further discussions. Thanks again!

---

> ### Author Response · Authors · 2025-11-24
> **Expecting to receive further feedback**
>
> Dear Reviewer Cwki,
>
> Thank you once again for your thoughtful and constructive feedback. We have carefully addressed each of your comments with detailed responses, conducted additional experiments, and provided more in-depth analysis based on your latest suggestions. We have also uploaded a revised version of the paper with all modifications clearly highlighted.
>
> Your insights have been invaluable in helping us strengthen our work, and we would greatly appreciate it if you could let us know whether there are any remaining issues or points that may require further clarification. We would be more than happy to provide any additional explanation you may need.
>
> Thank you again for your time and valuable input. We look forward to hearing from you.
>
> Best regards,
>
> The Authors

---

> ### Author Response · Authors · 2025-12-01
> **Appreciation from Authors**
>
> Dear Reviewer Cwki,
>
> Thank you sincerely for the time and effort you dedicated to reviewing our submission. We truly appreciate your thoughtful comments and valuable feedback. We regret that we were unable to engage in further discussion with you before the software bug was discovered, and we understand that you are now unable to modify your score or participate in the rebuttal phase.
>
> Nevertheless, we would like to kindly emphasize that we have carefully addressed each of the concerns you raised, provided detailed responses, and incorporated the necessary revisions into the updated version of the paper. Your feedback has been instrumental in helping us strengthen our work, and we are genuinely grateful for your contribution.
>
> Thank you once again for your support and for the valuable insights you shared with us.
>
> Warm regards,
>
> The Authors

---

### Official Review · Reviewer_MGLn · 2025-10-31

**Soundness:** 3
**Presentation:** 2
**Contribution:** 3
**Rating:** 4
**Confidence:** 3

**Summary:**

In sequential recommendation, the recommender system predicts the next item that the user is likely to interact with. In some cases, users also submit natural language queries while interacting with the platform (often search queries, e.g. "blue dresses" or "shower curtains"), but these queries are typically ignored by the sequential recommendation system. The goal of this work is to use these queries to adapt the outputs of the sequential recommender system. Since the recsys is typically already trained and fixed, rather than changing the underlying model, the authors propose a masking strategy. In particular, in their method, the request is encoded and then converted into a binary mask that is applied to either the output weights of the attention layers or the feed-forward weights of the transformer. The idea is simple (in a good way), though the paper’s current presentation makes it appear more complex than necessary and would benefit from clearer exposition.

**Strengths:**

Strengths:
- The masking idea is simple and straight-forward
- The method is a computationally efficient to adapt a fixed sequential transformer-based recommender system, and is on par in runtime to all baselines
- The authors test 5 methods (including theirs) on 4 datasets, and find that their method outperforms the baselines in all settings in terms of NDCG, MRR, HR

**Weaknesses:**

# Clarity
The paper’s clarity could be substantially improved. For example, the proposed masking method is actually quite straightforward: request-specific masks are learned to adapt the fixed transformer model, and these masks are trained end-to-end by maximizing the likelihood of recommending relevant items. However, I found the discussion of a variational lower bound in Section 3.2 confusing since the authors use Gumbel-top-k to sample binary masks. Typically, one would then use the straight-through estimator, not variational inference. And in fact, the method does ultimately use the straight-through estimator and does not perform variational inference. I suggest removing the discussion of variational inference, at least from the main text, as it adds unnecessary complexity without clear benefit. Similarly, the KL regularization term does not need to be motivated from a variational inference perspective.

# Experiments
- The authors state that in "45 out of 48 experiments, CRAMER showed statistically significant improvements over the second best result", but they do not perform any multiple testing corrections. Please update the paper with corrections for multiple testing.
- For the efficiency results in Table 2, which backbone is this with?  It would be helpful to expand the table to include results for both backbones, and to report the runtime of the vanilla SASRec and BERT4Rec backbones themselves. This would clarify how much of the runtime is attributable to the query-aware method versus the backbone.

# Limitations
The authors do not provide any discussion of limitations in the paper. Please update the paper with a discussion of limitations. Here are a few examples of topics to consider adding, though the authors should add their own points beyond what I list here:
- The authors say in the introduction that "natural-language requests may emphasize or even contradict historical
preferences, requiring the model to dynamically balance immediate intent with long-term behavior patterns". However, in practice, they train their model to predict historical interactions conditional on the query. This raise the question of how much the queries can steer the model to items that users have truly not engaged with in the past.
- Relatedly, the method requires the platform to already have a good way of catering to the user's query (e.g. by changing the pool of items that are received), otherwise the historical interactions used for training will not truly reflect the user's intent
- General limitations of transformer-based sequential models, e.g., support for extremely large item inventories and generalization to new items

**Questions:**

One of the key premises of the paper is that the sequential recommender system cannot be retrained to incorporate user queries, which serves as an important motivation for this work. However, I am unclear about how this aligns with real-world practice, where recommender systems are often used alongside search queries. For example, when a user searches for something like "cat food," is it typical for the system to first retrieve a set of relevant items and then constrain the sequential recommender to only select from within that set? If so, while the current paper focuses solely on the sequential recommender in isolation, this combined approach of constraining the item pool based on the query and then applying the recommender could serve as a practical and relevant baseline for comparison.

---

> ### Author Response · Authors · 2025-11-17
> **Rebuttal by Authors**
>
> Thanks for acknowledging our work and the constructive comments. **We have uploaded a revised version of our paper based on your comments and have highlighted all modifications.** Below, we provide point-by-point responses to the weaknesses and questions you raised. We kindly invite you to review them.
>
> 1. **Discussion of Section 3.2 (W-Clarity)**: We thank you for the helpful suggestion. Our use of a variational perspective in Section 3.2 was introduced as a theoretical motivation: the request-to-mask mechanism can be viewed through a probabilistic control lens, where sparse gating variables are endowed with a prior and optimized under a KL-regularized objective. This framing, inspired by probabilistic control theory, provides conceptual grounding for why structured, selective masking is an effective interface for controllable editing of frozen recommenders. For this reason, we believe the variational motivation remains valuable for understanding the design of CRAMER. At the same time, we fully agree that the practical optimization relies on Gumbel–Top-$k$ sampling with a straight-through estimator, rather than full variational inference. **To improve clarity, the revised version now clarifies this distinction in the main text and moves part of the explanation to the Appendix A to streamline the exposition while preserving the underlying rationale.** We hope these changes address your concern without losing the theoretical context that guided our design.
>
> 2. **Multiple testing corrections (W-Experiments 1)**: Our original analysis treated the 48 hypothesis tests independently, which indeed leads to an overly optimistic estimate of statistical significance. In response, we have applied the Benjamini–Hochberg (BH) procedure (FDR = 0.05) across all 48 paired t-tests. After correction, CRAMER remains statistically significantly better than the strongest baseline in 41 settings, demonstrating that the observed improvements are robust rather than artifacts of multiple comparisons. **We have updated Table 1’s caption and the corresponding discussion in Section 4.2 in the revised version to reflect this corrected analysis.**
>
> 3. **Model efficiency (W-Experiments 2)**: Thank you for the insightful suggestion. In the original version, we only reported efficiency results using SASRec as the backbone because we observed that the additional overhead introduced by each request-aware method is likewise comparable across the two backbones. Nonetheless, we agree that providing both backbones offers a clearer and more complete picture. **In the revised version, we have updated Table 2 to include the runtime and GPU memory of SASRec and BERT4Rec themselves, followed by the incremental overhead on both backbones introduced by each request-aware method. We also updated the corresponding explanation in Section 4.4 to reflect these additions.**
>
> 4. **Clarification of our motivation and scenario (W-Limitations 1)**: We do mention "contradict historical preferences" in Section 1, but our motivation is not that users overturn their entire preference profile, but that natural-language requests often express temporary, context-dependent constraints that override only specific aspects of their long-term preferences. For example, a user who usually prefers spicy, meaty, Asian dishes may issue a request like “My stomach hurts today; no spicy food.” Here the user negates only the spicy dimension while preserving the rest. At this moment, the user probably needs us to recommend some Asian dishes with plenty of meat, but not spicy ones. CRAMER is designed to support this selective, fine-grained modulation of a frozen backbone rather than steering users toward items unrelated to their overall tastes. **We have clarified this nuance in Section 1 of the revised version.**
>
> 5. **Requirement for platforms (W-Limitations 2)**: CRAMER does not assume that the platform already implements a complex method of catering to the user's request. In our intended use case, the platform only needs to provide a simple feedback box after generating a recommendation list, where the user can issue a natural-language request. CRAMER then interprets this request and refreshes the list immediately by modulating the frozen backbone, without needing additional retrieval mechanisms.
>
> 6. **General limitations of Transformer-based sequential recommenders (W-Limitations 3)**: We agree that CRAMER inherits the general limitations of Transformer-based sequential recommenders, including challenges in scaling to extremely large item inventories and limited generalization to entirely new or unseen items. These constraints arise from the backbone architecture rather than the proposed request-aware masking mechanism, but we are willing to mention them. **In the revised version, we have discussed them in the newly added Appendix C, along with other possible limitations.**

---

> ### Author Response · Authors · 2025-11-17
> **Rebuttal by Authors**
>
> 7. **How CRAMER aligns with real-world practice (Q)**: Our work targets a different but common interaction scenario. In many cases, users first receive a recommendation list (e.g., homepage recommendations without an explicit search intent), and only then feel the need to provide fine-grained natural-language feedback because their immediate intent may differ from what their historical behavior suggests. Such feedback is not a search query that filters the candidate pool, but an immediate correction applied on top of the sequential recommender’s output. CRAMER is designed specifically for this “post-recommendation request” setting, where the system must rapidly reinterpret the user’s intent and refresh the list without retraining or invoking a new retrieval stage. While retrieval-plus-ranking pipelines are valuable, they address a different problem from the one we focus on.
>
> We hope the above response can fully address your concerns and look forward to further discussions. Thanks again!

---

> ### Author Response · Authors · 2025-11-24
> **Expecting to receive further feedback**
>
> Dear Reviewer MGLn,
>
> Thank you once again for your thoughtful and constructive feedback. We have carefully addressed each of your comments with detailed responses, conducted additional experiments, and provided more in-depth analysis based on your latest suggestions. We have also uploaded a revised version of the paper with all modifications clearly highlighted.
>
> Your insights have been invaluable in helping us strengthen our work, and we would greatly appreciate it if you could let us know whether there are any remaining issues or points that may require further clarification. We would be more than happy to provide any additional explanation you may need.
>
> Thank you again for your time and valuable input. We look forward to hearing from you.
>
> Best regards,
>
> The Authors

---

> > ### Comment · Reviewer_MGLn · 2025-11-26
> >
> > I thank the authors for their detailed rebuttal. My concerns have been addressed, and I have raised my score.

---

> > > ### Author Response · Authors · 2025-11-26
> > > **Thanks and follow-up to reviewer MGLn**
> > >
> > > Dear Reviewer MGLn,
> > >
> > > Thank you very much for your positive response and for your kind recognition of our work. We are truly glad to hear that our rebuttal has addressed your concerns and that you are willing to raise your score. If there are any remaining questions or points that could benefit from further discussion, please feel free to let us know—we would be happy to provide any additional clarification.
> > >
> > > Once again, we sincerely appreciate your dedication to scientific rigor and your valuable help in improving our paper.
> > >
> > > Best regards,
> > >
> > > The Authors

---

### Official Review · Reviewer_VS3F · 2025-11-01

**Soundness:** 3
**Presentation:** 3
**Contribution:** 3
**Rating:** 4
**Confidence:** 4

**Summary:**

This paper presents CRAMER (Control via Request-Aware Masking for Editing Recommenders), a novel, parameter-efficient framework designed to enable sequential recommendation models to respond instantaneously to natural-language user requests. The core innovation is treating the user request as a control signal used to modulate the parameters of a frozen Transformer backbone via structured, sparse masking, drawing inspiration from model control theory. CRAMER avoids the computational overhead associated with full fine-tuning or heavyweight Large Language Model (LLM) inference. The methodology employs a lightweight request encoder, converts the semantic embedding into sparse row–column gate vectors using the Gumbel–Top-k trick, and trains this mapping using a variationally motivated objective featuring a KL sparsity regularizer. Empirically, the authors demonstrate that CRAMER consistently outperforms four state-of-the-art request-aware baselines across multiple large-scale benchmark datasets and two foundational Transformer architectures (SASRec and BERT4Rec)

**Strengths:**

1. Superior Empirical Performance: CRAMER demonstrates robust and consistent state-of-the-art performance, outperforming four competing request-aware baselines (Query-SeqRec, BLaIR, LLM-ESR, REARANK) across all four benchmark datasets (ReDial, KuaiSAR, Beauty, CDs&Vinyl) and both frozen backbones (SASRec and BERT4Rec). Notably, CRAMER achieved statistically significant improvements over the second-best result in 45 out of 48 experiments (93.75%).

2. Exceptional Efficiency and Scalability: The framework is highly efficient, achieving minimal computational overhead compared to retraining or LLM-based approaches. CRAMER’s average inference runtime is 0.018 seconds per request, which is comparable to the fastest baseline (LLM-ESR at 0.016s) but drastically faster than heavyweight methods like REARANK (9.256s), making it highly suitable for real-time recommendation deployment.

3. Novel and Parameter-Efficient Control Paradigm: CRAMER introduces a new paradigm for request-aware sequential recommendation by framing the natural language request as a fine-grained control signal applied via masking to a frozen backbone. This masking approach, utilizing structured row-column gating, is an expressive yet lightweight mechanism that minimizes the number of trainable parameters required for adaptation

**Weaknesses:**

1. Sensitivity to Mask Sparsity (ρ): The drop ratio ρ is a crucial hyperparameter, and performance deteriorates significantly at extreme values. Optimal performance requires tuning ρ based on dataset characteristics—smaller ρ (denser masks) is better for large, data-dense datasets, while larger ρ (sparser masks) is preferable for small datasets to prevent overfitting. This suggests a non-trivial tuning requirement for new application domains.

2. Dependence on High-Capacity PLMs: The performance of CRAMER is shown to rely heavily on the quality and capacity of the Pretrained Language Model (PLM) used for initialization of the request encoder $E_{ϕ_{enc}}$. While RoBERTa and ModernBERT yield the best results, lightweight models like MiniLM consistently underperform, indicating that robust semantic extraction is a prerequisite for effective control via masking.

3. Use of a Variationally Inspired Surrogate Objective: The authors acknowledge that the practical training objective is not a strict Evidence Lower Bound (ELBO) maximization. Because the forward pass uses a hard k-hot Gumbel–Top-k sampler, while the KL regularizer assumes independent Bernoulli gates, the resulting objective is a "variationally inspired surrogate". A deeper discussion or boundary analysis concerning the theoretical gap introduced by this necessary approximation would strengthen the paper.

4. Training Complexity of Discrete Optimization: The method relies on sophisticated techniques (Gumbel–Top-k sampling and Straight-Through Estimators (STE)) to handle discrete mask generation and gradient flow. Furthermore, the performance is sensitive to the annealing schedule for the STE temperature $\tau$, with cosine decay found to be superior. These elements add overhead and complexity to the training setup compared to fully continuous optimization methods.

5. Incomplete Large-Scale Validation: While the paper claims superiority on "large-scale" benchmarks, the three largest datasets used (KuaiSAR, Beauty, and CDs\&Vinyl) were randomly downsampled due to computing budget limitations. Although necessary for the authors, this compromises the ability to definitively evaluate the method's performance and scalability claims on the full, unreduced datasets.

**Questions:**

NA

---

> ### Author Response · Authors · 2025-11-17
> **Rebuttal by Authors**
>
> Thanks for acknowledging our work and the constructive comments. **We have uploaded a revised version of our paper based on your comments and have highlighted all modifications.** Below, we provide point-by-point responses to the weaknesses and questions you raised. We kindly invite you to review them.
>
> 1. **Sensitivity to $\rho$ (W1)**: While $\rho$ is an important hyperparameter, our experiments show that CRAMER is robust within a broad practical range. As illustrated in Figure 2, $\rho \in [0.05, 0.20]$ consistently delivers strong performance across all datasets, with only mild variation rather than sharp degradation. This allows those who use the model to rely on stable defaults (e.g., around 0.10 for large datasets;  0.15–0.20 for smaller ones) without extensive tuning. Moreover, CRAMER’s efficiency and lightweight training cost makes such tuning inexpensive in practice. Overall, CRAMER does not require heavy hyperparameter search to perform well in new domains.
>
> 2. **Dependence on high-capacity PLMs (W2)**: We thank you for pointing out this aspect. To verify whether the performance variation comes from CRAMER itself or from the intrinsic semantic ability of the PLM, we conducted an additional sensitivity study using four BERT variants of increasing capacity (Tiny-Mini-Medium-Base; Base is the default BERT used in the paper). Using SASRec as backbone, the NDCG@10 results across all datasets are:
>
> | PLM Variant | Tiny  | Mini  | Medium | Base  |
> | ----------- | ----- | ----- | ------ | ----- |
> | ReDial      | 0.403 | 0.411 | 0.419  | 0.428 |
> | KuaiSAR     | 0.420 | 0.425 | 0.432  | 0.436 |
> | Beauty      | 0.465 | 0.474 | 0.481  | 0.489 |
> | CDs&Vinyl   | 0.449 | 0.454 | 0.460  | 0.472 |
>
> In all cases, performance follows a strictly monotonic order that exactly matches PLM capacity (Tiny < Mini < Medium < Base). For a single dataset, the probability of observing such a perfect ordering by chance is $p=1/24 \approx 0.0417 < 0.05$, and seeing the same trend across all four datasets further supports that the variation arises from the intrinsic semantic capacity of the PLMs, not from instability in CRAMER. In practice, CRAMER simply leverages the cues provided by the request encoder—stronger PLMs naturally enable more precise control. This is not a limitation of our framework; in fact, even with classic models such as BERT we already achieve strong performance, and given the clear trend that future PLMs and semantic encoders will only become more capable and efficient, our approach is inherently forward-looking and poised to benefit from these advancements. **We have added the relevant analysis to the revised version; please refer to Appendix B.5.**
>
> 3. **Variationally inspired surrogate objective (W3)**: We appreciate your careful observation that our training objective is a variationally inspired surrogate rather than a strict ELBO. This type of design follows a standard practice in sparse gating and masking methods, where discrete structures are optimized via continuous relaxations and approximate objectives (e.g., Straight-Through Estimator \[1\], Concrete Relaxation \[2\], and $L_0$ Regularization \[4\]). In our setting, the KL term primarily acts as a sparsity-inducing regularizer over gates rather than an exact posterior matching term, and empirically we observe stable training and robust performance across datasets and backbones. **In the revised version, we have added a “Discussion on Surrogate Objective” paragraph in Appendix A explicitly discussing this approximation and its connection to prior work.**
>
> 4. **Training complexity (W4)**: While CRAMER employs Gumbel–Top-$k$ sampling and a straight-through estimator, the overall training pipeline remains lightweight because the backbone is frozen and the request-to-mask module is extremely small. These components follow well-established practice in sparse gating and masking methods \[1\]\[2\]\[3\]\[4\], and in practice introduce negligible computational overhead compared to fine-tuning or adapter-based approaches. Regarding temperature scheduling, CRAMER is not sensitive to the specific annealing strategy. As shown in Figure 8, cosine decay performs only marginally better in stability and final performance, while linear and exponential decay produce equally strong results and stability, indicating that the method does not rely on a brittle schedule.

---

> ### Author Response · Authors · 2025-11-17
> **Rebuttal by Authors**
>
> 5. **Large-scale validation (W5)**: Although we downsampled the largest datasets, we note that the original versions contain hundreds of thousands to millions of items, which far exceed the scale typically handled directly in sequential recommendation (where candidate generation or coarse ranking is applied in practice). Even after downsampling, our datasets still contain $4×10^4$-$2×10^5$ items (Table 3), which is already close to or larger than those used in existing recommendation papers such as SASRec ($4×10^4$), BERT4Rec ($5×10^4$), or BLaIR ($1.8×10^5$). We believe our settings provide a rigorous evaluation.
>
> We hope the above response can fully address your concerns and look forward to further discussions. Thanks again!
>
> ### References
>
> \[1\] Estimating or Propagating Gradients Through Stochastic Neurons for Conditional Computation, Bengio et al., 2013.
>
> \[2\] The Concrete Distribution: A Continuous Relaxation of Discrete Random Variables, Maddison et al., 2016.
>
> \[3\] Categorical Reparameterization with Gumbel-Softmax, Jang et al., 2016.
>
> \[4\] Learning Sparse Neural Networks through $L_0$ Regularization, Louizos et al., 2017.

---

> ### Author Response · Authors · 2025-11-24
> **Expecting to receive further feedback**
>
> Dear Reviewer VS3F,
>
> Thank you once again for your thoughtful and constructive feedback. We have carefully addressed each of your comments with detailed responses, conducted additional experiments, and provided more in-depth analysis based on your latest suggestions. We have also uploaded a revised version of the paper with all modifications clearly highlighted.
>
> Your insights have been invaluable in helping us strengthen our work, and we would greatly appreciate it if you could let us know whether there are any remaining issues or points that may require further clarification. We would be more than happy to provide any additional explanation you may need.
>
> Thank you again for your time and valuable input. We look forward to hearing from you.
>
> Best regards,
>
> The Authors

---

> ### Author Response · Authors · 2025-12-01
> **Appreciation from Authors**
>
> Dear Reviewer VS3F,
>
> Thank you sincerely for the time and effort you dedicated to reviewing our submission. We truly appreciate your thoughtful comments and valuable feedback. We regret that we were unable to engage in further discussion with you before the software bug was discovered, and we understand that you are now unable to modify your score or participate in the rebuttal phase.
>
> Nevertheless, we would like to kindly emphasize that we have carefully addressed each of the concerns you raised, provided detailed responses, and incorporated the necessary revisions into the updated version of the paper. Your feedback has been instrumental in helping us strengthen our work, and we are genuinely grateful for your contribution.
>
> Thank you once again for your support and for the valuable insights you shared with us.
>
> Warm regards,
>
> The Authors

---

### Author Response · Authors · 2025-12-01
**Summary Comment by Authors**

Dear **AC, SAC and PC**,

We sincerely thank all four reviewers for their thoughtful, constructive, and encouraging feedback. We are very pleased that **all reviewers consistently highlighted three core strengths of our work**:

1. **Novelty of the approach**: Each reviewer independently recognized the conceptual novelty of CRAMER. Reviewers described our masking-based control mechanism as **"novel and parameter-efficient paradigm" (VS3F)**, **"simple, straight-forward and in a good way" (MGLn)**, **"novel lightweight control method" (Cwki)**, **"novel theoretically grounded control mechanism" (cUQE)**, and fundamentally different from prior request-aware adaptation methods.
2. **Computational efficiency and ease of adaptation**: All reviewers agreed that CRAMER achieves real-time, parameter-efficient adaptation and resolves the efficiency–flexibility trade-off faced by existing methods. Multiple reviewers (VS3F, Cwki, cUQE) explicitly praised the **minimal overhead, frozen-backbone design, and plausibility for real-world deployment**.
3. **Robust empirical performance across datasets, backbones, and settings**: All reviewers agreed that our empirical results to be **consistently strong and robust**, noting CRAMER’s state-of-the-art performance in nearly all experiments, extensive ablations, cross-domain generalization, and stability across hyperparameters.

Across the four reviews, we were encouraged that weakness points raised by reviewers are actually constructive suggestions for further improvement, such as enhancing clarity, expanding runtime tables, and analyzing failure cases, which we have included in our revision during the rebuttal phase. We **implemented every suggested improvement**, added new experiments and analyses, and **submitted a fully revised version with clearly highlighted changes**.

Two reviewers (MGLn and cUQE) explicitly acknowledged that they were **very satisfied** with our response and their concerns had been **completely addressed**, with MGLn noting that they had already **raised their score (from 4 to 6)**. Therefore, we would like to emphasize that our final score was **6, 6, 6, 4 (average score: 5.5)** before the platform bug was discovered. We sincerely appreciate their time, detailed feedback, and willingness to acknowledge that the revisions strengthened the work.

For the remaining two reviewers (VS3F and Cwki), we fully understand that this year’s platform bug unfortunately prevented them from participating in the discussion or updating their reviews and scores. We nevertheless wish to express our sincere gratitude for their thoughtful comments, which substantially helped improve our paper. Our rebuttal and revision **carefully addressed every point they raised**, and the revised version reflects their suggestions throughout.

To summarize:
-  **All reviewers recognized the novelty, efficiency, robustness, and practical relevance of CRAMER.**
-  **We implemented all requested improvements and updated corresponding sections in the revised version**. These improvements include:
  1) statistical multiple-testing correction (MGLn)
  2) expanded runtime comparison (MGLn)
  3) clarified theoretical exposition (VS3F)
  4) additional PLM sensitivity analysis (VS3F, Cwki, cUQE)
  5) interpretability experiments (Cwki)
  6) analysis on ambiguous, contradictory and rare-term requests (Cwki, cUQE)
  7) discussion of limitations (MGLn, Cwki)
- **Reviewers who joined the discussion confirmed full satisfaction with our responses and revisions.**

We are grateful for the reviewers' positive assessments and constructive insights, which significantly strengthened the final version of our paper. Thank you again for your time, effort, and valuable contributions.

Sincerely,

The Authors

---

### Meta-Review · Area_Chair_6HpD · 2026-01-11

**Summary:**

This paper proposes CRAMER, a request-aware masking framework that modulates frozen sequential recommender models via natural-language user requests to achieve real-time adaptation without retraining. All reviewers recognized its efficiency, while raising constructive concerns about its practical limitations and experimental completeness. The editing nature with diverse user requests might hurt the real-world performance.  Besides, user-controllable recommendation is a traditional task. Existing user-controllable recommendation work should be deeply discussed since there are more technical solutions to achieve request-aware recommendation.

**Reviewer Concerns:**

Addressed Concerns:
- Dependence on high-capacity PLMs: Conducted PLM sensitivity analysis
- Insufficient clarity in variational objective exposition: Clarified
- Incomplete experimental validation: Tested with further experiments.

Outstanding Concerns:
- Ambiguity/contradiction in user requests, partly validated.
- Lack of mask interpretability: Added experimental analysis to demonstrate mask-semantic mapping via request-type impact on recommendation results. However, how this mask maps to the disentangled parameters is unclear.

**Reviewer Scores:**

- Reviewer VS3F (Original 4): Core concerns partly addressed, score likely to rise or maintain.
- Reviewer MGLn (Original 4): Concerns partly addressed, score likely to rise or maintain.
- Reviewer Cwki (Original 6): Core concerns addressed, score likely to remain stable.
- Reviewer cUQE (Original 6): Concerns fully addressed, maintained original score.

---

### Decision · Program_Chairs · 2026-01-26

Reject